# GOcats: A tool for categorizing Gene Ontology into subgraphs of user-defined concepts

**Eugene W. Hinderer, III**[1]**, Hunter N. B. Moseley**[1,2,3,4,5]*

**1** Department of Molecular and Cellular Biochemistry, University of Kentucky, Lexington, Kentucky, United States of America, **2** Markey Cancer Center, University of Kentucky, Lexington, Kentucky, United States of America, **3** Resource Center for Stable Isotope-Resolved Metabolomics, University of Kentucky, Lexington, Kentucky, United States of America, **4** Institute for Biomedical Informatics, University of Kentucky, Lexington, Kentucky, United States of America, **5** Center for Clinical and Translational Science, University of Kentucky, Lexington, Kentucky, United States of America

* hunter.moseley@uky.edu

**Data Availability Statement:** GOcats is an open-source Python software package under a BSD-3 License, available on GitHub at https://github.com/ MoseleyBioinformaticsLab/GOcats and on the

## Abstract

Gene Ontology is used extensively in scientific knowledgebases and repositories to organize a wealth of biological information. However, interpreting annotations derived from differential gene lists is often difficult without manually sorting into higher-order categories. To address these issues, we present GOcats, a novel tool that organizes the Gene Ontology (GO) into subgraphs representing user-defined concepts, while ensuring that all appropriate relations are congruent with respect to scoping semantics. We tested GOcats performance using subcellular location categories to mine annotations from GO-utilizing knowledgebases and evaluated their accuracy against immunohistochemistry datasets in the Human Protein Atlas (HPA). In comparison to term categorizations generated from UniProt's controlled vocabulary and from GO slims via OWLTools' Map2Slim, GOcats outperformed these methods in its ability to mimic human-categorized GO term sets. Unlike the other methods, GOcats relies only on an input of basic keywords from the user (e.g. biologist), not a manually compiled or static set of top-level GO terms. Additionally, by identifying and properly defining relations with respect to semantic scope, GOcats can utilize the traditionally problematic relation, has_part, without encountering erroneous term mapping. We applied GOcats in the comparison of HPA-sourced knowledgebase annotations to experimentally-derived annotations provided by HPA directly. During the comparison, GOcats improved correspondence between the annotation sources by adjusting semantic granularity. GOcats enables the creation of custom, GO slim-like filters to map fine-grained gene annotations from gene annotation files to general subcellular compartments without needing to hand-select a set of GO terms for categorization. Moreover, GOcats can customize the level of semantic specificity for annotation categories. Furthermore, GOcats enables a safe and more comprehensive semantic scoping utilization of go-core, allowing for a more complete utilization of information available in GO. Together, these improvements can impact a variety of GO knowledgebase data mining use-cases as well as knowledgebase curation and quality control.

Python Package Index (PyPI) at https://pypi.python.org/pypi/GOcats. Documentation can be found at http://gocats.readthedocs.io/en/latest/. The exact version of GOcats used in this study, along with all scripts used to generate results can be found in the Figshare repository at https://doi.org/10.6084/m9.figshare.7064516 and at https://doi.org/10.6084/m9.figshare.7064549. The version of GO used to generate these results is go-core (go.obo) data-version: releases/2016-01-12. The UniProt Controlled Vocabulary file can be found at https://www.uniprot.org/docs/subcell.txt. Associated GO terms are indicated in by the GO identifier in each stanza. Map2slim is available on GitHub (https://github.com/owlcollab/owltools/wiki/Map2Slim) and requires OWL Tools, also available via GitHub (https://github.com/owlcollab/owltools/wiki/Install-OWLTools#building-from-source). Subcellular location data was obtained from version 15 of the Human Protein Atlas and can be downloaded at http://v15.proteinatlas.org/download/subcellular_location.csv.zip.

**Funding:** This work was supported in part by grants NSF 1419282 (Moseley), NIH 1U24DK097215-01A1 (Higashi, Fan, Lane, Moseley), and NIH UL1TR001998-01 (Kern).

**Competing interests:** The authors have declared that no competing interests exist.

# Introduction

## Gene Ontology (GO)

The Gene Ontology (GO) [1] is the most common biology-focused controlled vocabulary (CV) used to represent information and knowledge distilled from most biological and biomedical research data generated today, from classic wet-bench experiments to high-throughput analytical platforms, especially omics technologies. Each CV term in GO is assigned a unique alphanumeric code and is used to annotate genes and gene products in many other databases, including UniProt [2] and Ensembl [3]. GO is divided into three sub-ontologies: Cellular Component, Molecular Function, and Biological Process. A graph represents each sub-ontology, where individual GO terms are nodes connected by directional edges (i.e. relation). For example, the term "lobed nucleus" (GO:0098537) is connected by a directional is_a relation edge to the term "nucleus" (GO:0005634). In this graph context, the is_a relation defines the term "nucleus" as a parent of the term "lobed nucleus". There are eleven types of relations used in the core version of GO; however, is_a is the most ubiquitous. The three GO sub-ontologies are "is_a disjoint" meaning that there are no is_a relations connecting any node among the three sub-ontologies.

There are also three versions of the GO database: *go-basic* which is filtered to only include is_a and part_of relations; *go* or *go-core* contains additional relations, that may span sub-ontologies and which point both toward and away from the top of the ontology; and *go-plus* contains cross-references to entries in external databases and ontologies.

## Growth and evolution of biological controlled vocabularies

GO and other CVs like the Unified Medical Language System [4,5] saw an explosion in development in the mid-1990s and early 2000s, coinciding with the increase in high-throughput experimentation and "big data" projects like the Human Genome Project. Their intended purpose is to standardize the functional descriptions of biological entities so that these functions can be referenced via annotations across large databases unambiguously, consistently, and with increased automation. However, ontology annotations are also utilized alongside automated pipelines that analyze protein-protein interaction networks and form predictions of unknown protein function based on these networks [6,7], for gene annotation enrichment analyses, and are now being leveraged for the creation of predictive disease models in the scope of systems biochemistry [8].

## Difficulty in representing biological concepts derived from omics-level research

Differential abundance analyses for a range of omics-level technologies, especially transcriptomics technologies can yield large lists of differential genes, gene-products, or gene variants. Many different GO annotation terms may be associated with these differential gene lists, making it difficult to interpret without manually sorting into appropriate descriptive categories [9]. It is similarly non-trivial to give a broad overview of a gene set or make queries for genes with annotations for a specific biological concept. For example, a recent effort to create a protein-protein interaction network analysis database resorted to manually building a hierarchical localization tree from GO cellular compartment terms due to the "incongruity in the resolution of localization data" in various source databases and the fact that no published method existed at that time for the automated organization of such terms [6]. If a subgraph of GO could be programmatically extracted to represent a specific biological concept, a category-

defining general term could be easily associated with all its ontological child terms within the subgraph.

Meanwhile, high-throughput transcriptomic and proteomic characterization efforts like those carried out by the Human Protein Atlas (HPA) now provide sophisticated pipelines for resolving expression profiles at organ, tissue, cellular and subcellular levels by integrating quantitative transcriptomics with microarray-based immunohistochemistry [10]. Such efforts create a huge amount of omics-level experimental data that is cross-validated and distilled into systems-level annotations linking genes, proteins, biochemical pathways, and disease phenotypes across our knowledgebases. However, annotations provided by such efforts may vary in terms of granularity, annotation sets used, or ontologies used. Therefore, (semi-)automated (i.e. at least partially automated) and unbiased methods for categorizing semantically-similar and biologically-related annotations are needed for integrating information from heterogeneous sources—even if the annotation terms themselves are standardized—to facilitate effective downstream systems-level analyses and integrated network-based modeling.

## Term categorization approaches

Issues of term organization and term filtering have led to the development of GO slims—manually trimmed versions of the gene ontology containing only generalized terms [11], which represent concepts within GO. Other software, like Categorizer [9], can organize the rest of GO into representative categories using semantic similarity measurements between GO terms. GO slims may be used in conjunction with mapping tools, such as OWLTools' (https://github.com/owlcollab/owltools) Map2Slim (M2S) or GOATools (https://zenodo.org/record/31628), to map fine-grained annotations within Gene Annotation Files (GAFs) to the appropriate generalized term(s) within the GO slim or within a list of GO terms of interest. While web-based tools such as QuickGO exist to help compile lists of GO terms [12], using M2S either relies completely on the structure of existing GO slims or requires input or selection of individual GO identifiers for added customization, and necessitates the use of other tools for mapping. UniProt has also developed a manually-created mapping of GO to a hierarchy of biologically-relevant concepts [13]. However, it is smaller and less maintained than GO slims, and is intended for use only within UniProt's native data structure.

## Semantic similarity in the context of broad term categorization

In addition to utilizing the inherent hierarchical organization of GO to categorize terms, other metrics may be used for categorization. For instance, semantic similarity can be combined along with the GO structure to calculate a statistical value indicating whether a term should belong to a predefined group or category of [9,14–17]. One rationale for this type of approach is that the topological distance between two terms in the ontology graph is not necessarily proportional to the semantic closeness in meaning between those terms, and semantic similarity reconciles potential inconsistencies between semantic closeness and graph distance. Additionally, some nodes have multiple parents, where one parent is more closely related to the child than the others [9]. Semantic similarity can help determine which parent is semantically more closely related to the term in question. While these issues are valid, we maintain that in the context of aggregating fine-grained terms into general categories, these considerations are not necessary. First, fluctuations in semantic distances between individual terms are not an issue once terms are binned into categories: all binned terms will be reduced to a single step away from the category-defining node. Second, the problem of choosing the most appropriate parent term for a GO term only causes problems when selecting a representative node for a category; however, since most paths eventually converge onto a common ancestor, any

significantly diverging paths would have its meaning captured by rooting multiple categories to a single term, cleanly sidestepping the issue.

## Maintenance of ontologies

Despite maintenance and standard policies for adding terms, ontological organization is still subject to human error and disagreement, necessitating quality assurance and revising, especially as ontologies evolve or merge. A recent review of current methods for biomedical ontology mapping highlights the importance in developing semi-automatic methods [18,19] to aid in ontology evolution efforts and reiterates the aforementioned concept of semantic correspondence in terms of scoping between terms [20]. Methods incorporating such correspondences have been published elsewhere, but these deal with issues of ontology evolution and merging, and not with categorizing terms into user-defined subsets [21,22]. Ontology merging also continues to be an active area of development for integrating functional, locational, and phenotypic information. To aid in this, another recent review points out the importance of integrating phenotypic information across various levels of organismal complexity, from the cellular level to the organ system level [8]. Thus, organizing location-relevant ontology terms into discrete categories is an important step toward this end.

## GO Categorization Suite (GOcats)

For the reasons indicated above, we have developed a tool called the GO Categorization Suite (GOcats), which serves to streamline the process of slicing the ontology into subgraphs representing specific biological concepts. Unlike previously developed tools, GOcats works with a list of user-provided keywords and/or GO terms, along with the structure of GO and augmented relation properties. Based on this input, GOcats automatically extracts a subgraph of related GO terms, identifies a representative category-defining GO term for the subgraph, and maps all subgraph child GO terms to this representative GO term. In essence, GOcats automatically generates a concept-specific GOslim with only keywords and GO terms provided by a user, typically a biologist. Furthermore, GOcats allows the user to choose between the strict axiomatic interpretation or a looser semantic scoping interpretation of part-whole (mereological) relation edges within GO. Specifically, we consider scoping relations to be comprised of is_a, part_of, and has_part, and mereological relations to be comprised of part_of and has_-part. In the next section, we evaluate GOcats ability to generate category-specific subgraphs and to utilize these subgraphs to compare knowledgebase annotations to their experimental source (i.e. the HPA). Due to the nature of the experimentally verified properties available from the HPA, our analysis in this paper focuses on cellular locations, especially subcellular locations. Also, this paper provides an in-depth description of GOcats's methods and their implementation. In a prior publication, we demonstrated GOcats's ability to improve gene-annotation enrichment analyses, involving all GO sub-ontologies [23].

## Results

### GOcats compactly organizes GO subcellular localization terms into user-specified categories

As an initial proof-of-concept, we evaluated the automatic extraction and categorization of 25 subcellular locations, using GOcats' "comprehensive" method of subgraph extension (See Methods and the go-core graph, data-version: releases/2016-01-12). Starting with common biological subcellular concepts like "nucleus", "cytoplasm", and "mitochondrion", we recursively used terms not being categorized to identify additional subcellular concepts and associated

keywords represented within the GO Cellular Component sub-ontology. Due to the eventual application to the HPA datasets, three unusual categories, "bacterial", "viral", and "other organism", were included to prevent categorization of terms that would complicate a eukaryotic interpretation of the other 22 subcellular locations, within the context of a greedy subgraph extension algorithm. For these resulting 25 categories, 22 contained a designated GO term root-node that exactly matched the concept intended at the creation of the keyword list (Table 1).

These subgraphs account for approximately 89% of GO's Cellular Component sub-ontology. While keyword querying of GO provided an initial seeding of the growing subgraph, Table 1 highlights the necessity of re-analyzing the GO graph, both to remove terms erroneously added by the keyword search and to add appropriate subgraph terms not captured by the keyword search. For example, the "cytoplasm" subgraph grew from its initial seeding of 296 nodes to 1197 nodes after extension. Conversely, 136 nodes were seeded by keyword for the "bacterial" subgraph, but only 16 were rooted to the representative node.

To assess the relative size and structure of subgraphs within GO, we visualized the category subgraphs as a network using Cytoscape 3.0 [24]. GOcats outputs a dictionary of individual GO term keys with a list of category-defining root-node values as part of its normal functionality.

Of note, 2102 of the 3877 terms in Cellular Component could be rooted to a single concept: "macromolecular complex." Despite cytosol being defined as "the part of the cytoplasm that does not contain organelles, but which does contain other particulate matter, such as protein complexes", less than half of the terms rooted to macromolecular complex also rooted to cytosol or cytoplasm. Surprisingly, approximately 25% of the terms rooted to macromolecular complex are rooted to this category alone (Fig 1A). In this visualization, intracellular organelles tend to be clustered about cytoplasm, except for nucleus which the GO consortium does not consider as part of the cytoplasm. The visualization of the subgraph contents confirmed the uniqueness of the macromolecular complex category and showed the relative sizes of groups of GO terms shared between two or more categories. But the macromolecular complex category somewhat complicates the visualization of category organization within GO, due to this category's size and interconnectedness within the ontology.

To better reflect what might be a biologist's expectation for a cell's overall organization, we produced another visualization with the macromolecular complex category omitted (Fig 1B). Despite the idiosyncrasies with the macromolecular complex subgraph, compartments that typically contain a large range of protein complexes, such as the nucleus, plasma membrane, and cytoplasm appear to be appropriately populated. Furthermore, concepts such as endomembrane trafficking can be gleaned from the network connectedness of representative nodes, such as lysosome, Golgi apparatus, vesicle, secretory granule, and cytoplasm. Overall, the patterns of connectedness in this network make more sense biologically, within the constraints of GO's internal organization. In other words, it is easier to see the expected biological relationships between cellular locations in Fig 1B versus Fig 1A.

## GOcats-derived category subgraphs compare well with similar subgraphs derived by other methods

We compared GOcats' category subgraphs taken from the go-core database, data-version: releases/2016-01-12 to subgraphs of the manually-curated UniProt subcellular localization controlled vocabulary (CV) [13] (see Fig 2 and Methods) and to subgraphs created by M2S (see Methods). Differences in the sets of GO terms contained within these subgraphs can be attributed to differences in the number of edges between nodes—as is the case between GOcats and M2S since M2S does not traverse across has_part edges—and the number of overall nodes

**Table 1. Summary of 25 example subcellular locations extracted by GOcats.**

| Subgraph name | User-input keywords | Predicted representative term (ID) | Nodes seeded from keyword search | Nodes added during graph extension | Seeded nodes not in subgraph | Total nodes |
|---|---|---|---|---|---|---|
| Aggresome | aggresome, aggresomal, aggresomes | aggresome (GO:0016235) | 1 | 0 | 0 | 1 |
| Bacterial | bacterial, bacteria, bacterial-type | bacterial-type flagellum (GO:0009288) | 136 | 1 | 121 | 16 |
| Cell Junction | junction | Cell junction (GO:0030054) | 68 | 16 | 34 | 50 |
| Chromosome | chromosome, chromosomal, chromosomes | chromosome (GO:0005694) | 120 | 122 | 31 | 211 |
| Cytoplasm | cytoplasm, cytoplasmic | Cytoplasm (GO:0005737) | 296 | 1061 | 160 | 1197 |
| Cytoplasmic Granule | granule, granules | secretory granule (GO:0030141) | 81 | 16 | 50 | 47 |
| Cytoskeleton | cytoskeleton, cytoskeletal | cytoskeleton (GO:0005856) | 78 | 194 | 47 | 225 |
| Cytosol | cytosol, cytosolic | cytosol (GO:0005829) | 56 | 51 | 28 | 79 |
| Endoplasmic Reticulum | endoplasmic, sarcoplasmic, reticulum | endoplasmic reticulum (GO:0005783) | 113 | 39 | 51 | 101 |
| Endosome | endosome, endosomes, endosomal | endosome (GO:0005768) | 67 | 15 | 24 | 58 |
| Extracellular | extracellular, secreted | extracellular region (GO:0005576) | 142 | 123 | 85 | 180 |
| Golgi Apparatus | golgi | golgi apparatus (GO:0005794) | 67 | 12 | 25 | 54 |
| Lysosome | lysosome, lysosomal, lysosomes | lysosome (GO:0005764) | 42 | 7 | 16 | 33 |
| Macromolecular Complex | protein, macromolecular | macromolecular complex (GO:0032991) | 1317 | 969 | 184 | 2102 |
| Microbody | microbody, microbodies | microbody (GO:0042579) | 4 | 20 | 0 | 24 |
| Mitochondrion | mitochondria, mitochondrial, mitochondrion | mitochondrion (GO:0005739) | 134 | 2 | 44 | 92 |
| Neuron Part | neuron, neuronal, neurons, synapse | neuron part (GO:0097458) | 90 | 94 | 35 | 149 |
| Nucleolus | nucleolus, nucleolar | nucleolus (GO:0005730) | 25 | 11 | 12 | 24 |
| Nucleus | nucleus, nuclei, nuclear | nucleus (GO:0005634) | 288 | 340 | 118 | 510 |
| Other Organism | other, host, organism | other organism (GO:0044215) | 369 | 12 | 259 | 122 |
| Plasma Membrane | plasma | plasma membrane (GO:0005886) | 308 | 302 | 164 | 446 |
| Plastid | plastid, chloroplast | plastid (GO:0009536) | 95 | 48 | 8 | 135 |
| Thylakoid | thylakoid, thylakoids | thylakoid (GO:0009579) | 52 | 22 | 11 | 63 |
| Vesicle | vesicle, vesicles | vesicle (GO:0031982) | 198 | 90 | 85 | 203 |
| Viral | virion, virus, viral | viral occlusion body (GO:0039679) | 93 | 1 | 26 | 68 |
| | Expected representative | | | | | |
| | Unexpected representative | | | | | |

[a]Nodes seeded from keyword search.

[b]Nodes added through subgraph extension.

[c]Seeded nodes removed due to subgraph omission.

[d]Because subgraph nodes may root to more than one representative root node, the totals in this table do not add up to the total number of GO terms in Cellular Component.

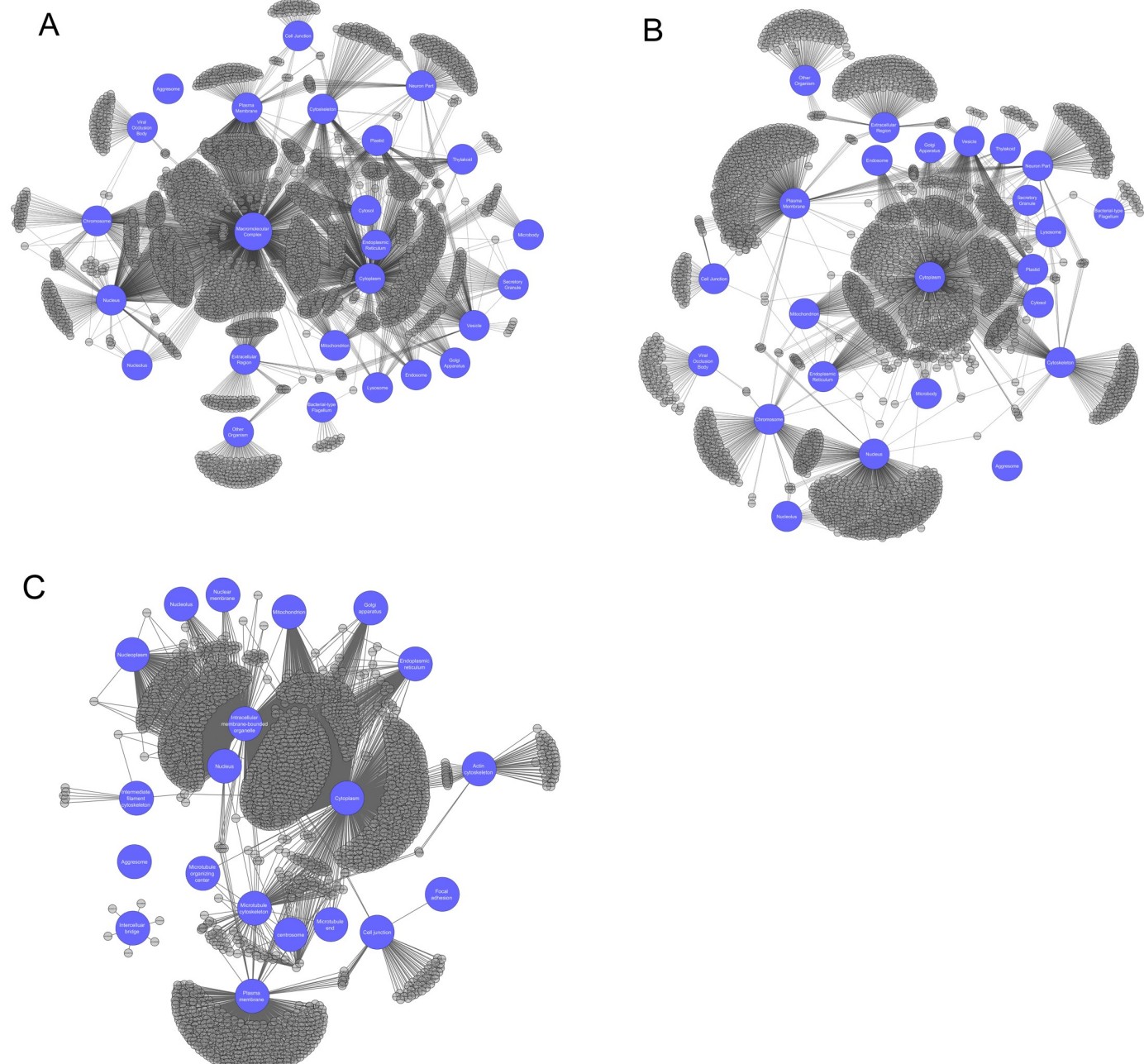

**Fig 1. A.** Network of 25 categories whose subgraphs account for 89% of the GO cellular component sub-ontology. **B.** Network of all categories from A except for Macromolecular Complex. **C.** Network of 20 categories used in the Human Protein Atlas subcellular localization immunohistochemistry raw data.

being evaluated—as is the case when comparing M2S and GOcats term sets to the UniProt CV terms sets since the UniProt CV contains considerably fewer GO terms. For the most part, GOcats category subgraphs are large supersets of UniProt CV subgraphs, as demonstrated by the high inclusion indices and low Jaccard indices in Table 2. In the comparison of GOcats and M2S subgraphs, the mappings for most categories are in very close agreement, as evidenced by both high inclusion and Jaccard indices in Table 3 and further highlighted in Fig 3A

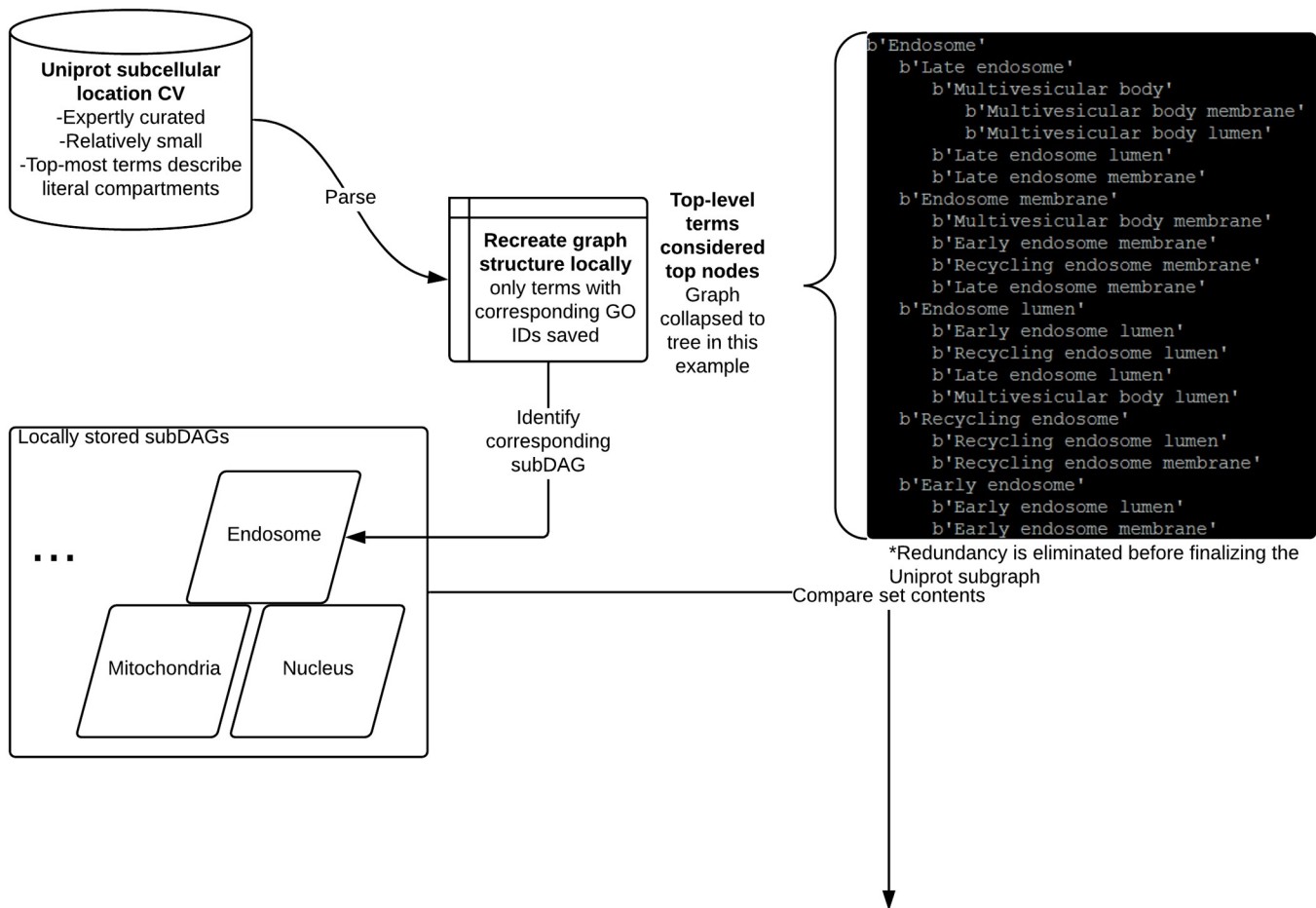

**Fig 2. Flowchart of the UniProt subcellular location CV subgraph creation method and inclusion index equation.**

and 3B and S1 Data A-V [25]. Overall, GOcats robustly categorizes GO terms into category subgraphs with high similarity to existing GO-utilizing categorization methods while including information gleaned from has_part edges.

However, in some categories, M2S and GOcats disagree as illustrated in Fig 3C and S1(E) Data. The most striking example of this is in the plasma membrane category, where M2S's subgraph contained over 300 terms that were not mapped by GOcats. We manually examined these discrepancies in the plasma membrane category and noted that many of the terms uniquely mapped by M2S did not appear to be properly rooted to "plasma membrane" (S2 Data). M2S mapped terms such as "nuclear envelope," "endomembrane system," "cell projection cytoplasm", and "synaptic vesicle, resting pool" to the plasma membrane category, while such questionable associations were not made using GOcats. Even though most terms

**Table 2. Agreement summary between corresponding GOcats and UniProt CV subgraphs.**

| Location Category | Term ID | Inclusion Index | Jaccard Index | GOcats subgraph size | UniProt CV subgraph size |
|---|---|---|---|---|---|
| Bacterial-type Flagellum | GO:0009288 | 1 | 0.0625 | 16 | 1 |
| Cell Junction | GO:0030054 | 0.47619 | 0.163934 | 50 | 21 |
| Chromosome | GO:0005694 | 1 | 0.0189573 | 211 | 4 |
| Cytoplasm | GO:0005737 | 0.809524 | 0.0141549 | 1197 | 21 |
| Endoplasmic Reticulum | GO:0005783 | 0.818182 | 0.0873786 | 101 | 11 |
| Endosome | GO:0005783 | 1 | 0.241379 | 58 | 14 |
| Extracellular Region | GO:0005576 | 0.5625 | 0.0481283 | 180 | 16 |
| Golgi Apparatus | GO:0005794 | 0.8 | 0.142857 | 54 | 10 |
| Lysosome | GO:0005764 | 1 | 0.0909091 | 33 | 3 |
| Mitochondrion | GO:0005739 | 1 | 0.0978261 | 92 | 9 |
| Nucleus | GO:0005634 | 1 | 0.0294118 | 510 | 15 |
| Plastid | GO:0009536 | 0.846154 | 0.307692 | 135 | 52 |

**Table 3. Agreement summary between corresponding GOcats and Map2Slim subgraphs.**

| Location Category | Term ID | Inclusion Index[‡] | Jaccard Index | GOcats subgraph size | Map2Slim subgraph size | "Has_part" relationships |
|---|---|---|---|---|---|---|
| Aggresome | GO:0016235 | 1 | 1 | 1 | 1 | 0 |
| Bacterial-type Flagellum | GO:0009288 | 1 | 1 | 16 | 16 | 8 |
| Cell Junction | GO:0030054 | 0.980392 | 0.980392 | 50 | 51 | 4 |
| Chromosome | GO:0005694 | 0.984375 | 0.883178 | 211 | 192 | 40 |
| Cytoplasm | GO:0005737 | 0.927273 | 0.452055 | 1197 | 605 | 38 |
| Cytoskeleton | GO:0005856 | 0.812274 | 0.812274 | 225 | 277 | 10 |
| Cytosol | GO:0005829 | 0.963415 | 0.963415 | 79 | 82 | 8 |
| Endoplasmic Reticulum | GO:0005783 | 1 | 0.990099 | 101 | 100 | 4 |
| Endosome | GO:0005768 | 1 | 1 | 58 | 58 | 0 |
| Extracellular Region | GO:0005576 | 1 | 0.927778 | 180 | 167 | 2 |
| Golgi Apparatus | GO:0005794 | 1 | 1 | 54 | 54 | 0 |
| Lysosome | GO:0005764 | 1 | 1 | 33 | 33 | 0 |
| Macromolecular Complex | GO:0032991 | 0.947274 | 0.947274 | 2102 | 2219 | 232 |
| Microbody | GO:0042579 | 1 | 1 | 2 | 24 | 0 |
| Mitochondrion | GO:0005739 | 0.978723 | 0.978723 | 92 | 94 | 8 |
| Neuron Part | GO:0097458 | 1 | 0.993289 | 149 | 148 | 22 |
| Nucleolus | GO:0005730 | 0.857143 | 0.857143 | 24 | 28 | 0 |
| Nucleus | GO:0005634 | 0.991684 | 0.928016 | 510 | 481 | 168 |
| Other Organism | GO:0044215 | 1 | 1 | 122 | 122 | 8 |
| Plasma Membrane | GO:0005886 | 0.563081 | 0.547097 | 446 | 753 | 20 |
| Plastid | GO:0009536 | 0.992647 | 0.992647 | 135 | 136 | 0 |
| Secretory Granule | GO:0030141 | 1 | 1 | 47 | 47 | 0 |
| Thylakoid | GO:0009579 | 1 | 1 | 63 | 63 | 0 |
| Vesicle | GO:0031982 | 0.981132 | 0.757282 | 203 | 159 | 12 |
| Viral Occlusion Body | GO:0039679 | 1 | 0.0147059 | 68 | 1 | 4 |

[‡] Inclusion index quantifies the extent to which the smaller subgraph is included in the larger subgraph

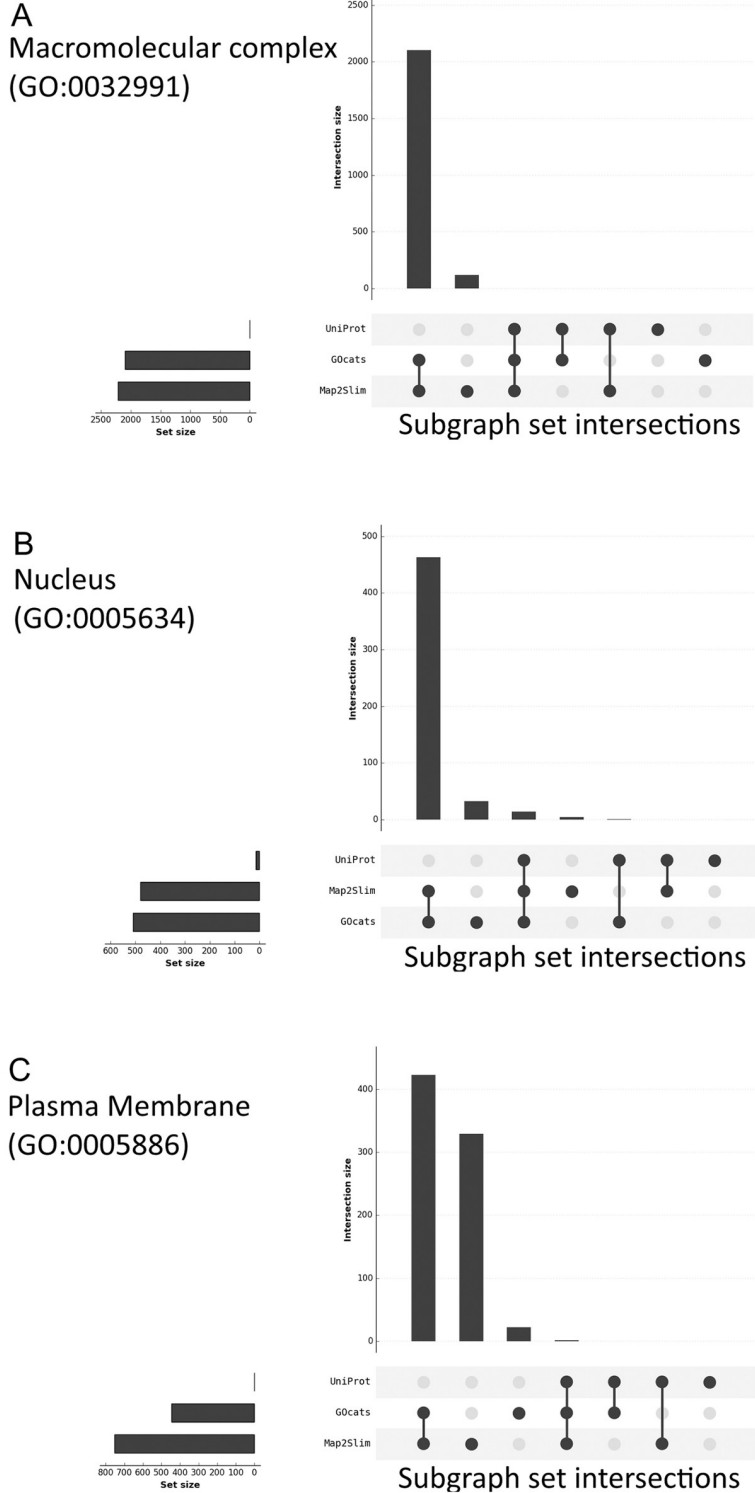

**Fig 3. Visualizing the degree of overlap between the category subgraphs created by GOcats, Map2Slim, and the UniProt CV.** Plots were created using the R package UpSetR [25], as a visual alternative to a Venn diagram. The amount of overlap between category-specific subgraphs are indicated by the vertical bar graph with the connected dots identifying which specific mapping method (UniProt, GOcats, and Map2Slim) is included in the overlap. A) Macromolecular Complex; B) Nucleus; C) Plasma Membrane. Plots for all categories can be found in S1(A-Y) Data.

included by M2S but excluded by GOcats exist beyond the scope of or are largely unrelated to the concept of "plasma membrane," a few terms in the set did seem appropriate, such as "intrinsic component of external side of cell outer membrane." However, of these examples, no logical semantic path could be traced between the term and "plasma membrane" in GO, indicating that these associations are not present in the ontology itself. **These differences in mapping are due to our reevaluation of the has_part edges with respect to scope**. As shown in Table 3 the categories with the greatest agreement between the two methods were those with no instances of has_part relations, which is the only relation in Cellular Component that is natively incongruent with respect to scope. However, there is no apparent correlation between the frequency of this relation and the extent of disagreement.

## Custom-tailoring of GO slim-like categories with GOcats allows for robust knowledgebase gene annotation mining

The ability to query knowledgebases for genes and gene products related to a set of general concepts-of-interest is an important method for biologists and bioinformaticians alike. We hypothesized that grouping annotations into categories using GOcats and relevant keywords would more closely match the annotations categorized manually by the HPA consortium than either M2S or UniProt's CV. Using the set of GO terms annotated in the HPA's immunohistochemistry localization raw data as "concepts" (Table 4), we derived mappings to annotation categories generated from GOcats, M2S, and UniProt's CV based on UniProt- and Ensembl-sourced annotations from the European Molecular Biology Laboratories-European Bioinformatics Institute (EMBL-EBI) QuickGO knowledgebase resource [12] (See Methods). In this context, the term "raw data" refers to processed, curated experimental data that is annotated as a contrast to the GO annotations derived from a knowledgebase.

Next, we evaluated how these derived annotation categories matched raw HPA data GO annotations (See Fig 4 and Methods). GOcats slightly outperformed M2S and significantly outperformed UniProt's CV in the ability to query and extract genes and gene products from the knowledgebase that exactly matched the annotations provided by the HPA (Fig 5A). Similar relative results are seen for partially matched knowledgebase annotations. Genes in the "partial agreement," "partial agreement is superset," or "no agreement" groups may have annotations from other sources that place the gene in a location not tested by the HPA immunohistochemistry experiments or may be due to non-HPA annotations being at a higher semantic scoping than what the HPA provided. Also, novel localization provided by the HPA could explain genes in the "partial agreement" and "no agreement" groups. In this context, "partial agreement" refers to genes with at least one matching subcellular location, "partial agreement is superset" refers to genes where knowledgebase subcellular locations are a superset of the HPA dataset (these are mutually exclusive to the "partial agreement" category), "no agreement" refers to genes with no subcellular locations in common, and "no annotations" refers to genes in the experimental dataset that were not found in the knowledgebase.

Furthermore, GOcats performed the categorization of HPA's subcellular locations dataset in an average of 10.574 seconds after 50 test runs (standard deviation of 0.074 seconds), while M2S performed its mapping on the same data in an average of 14.837 seconds after 50 test runs (standard deviation of 0.300 seconds) (see Methods for hardware configuration details). These results are rather surprising since GOcats is implemented in Python [26], an interpreted language, versus M2S which is implemented in Java and compiled to Java byte code. However, through the use of Python decorators, GOcats recursively creates and stores ancestor and descendent node sets in a manner analogous to lazy evaluation, allowing the implementation of efficient subgraph-centric algorithms that only precomputes the ancestor and descendent

**Table 4. Summary of 20 subcellular locations used in the HPA raw experimental data extracted by GOcats.**

| Subgraph name | User-input keywords | Predicted representative term (ID) | Nodes seeded from keyword search | Nodes added during graph extension | Seeded nodes not in subgraph[a] | Total nodes[b] |
|---|---|---|---|---|---|---|
| Actin cytoskeleton | actin cytoskeleton | actin cytoskeleton (GO:0015629) | 117 | 22 | 77 | 62 |
| Aggresome | aggresome, aggresomal, aggresomes | aggresome (GO:0016235) | 1 | 0 | 0 | 1 |
| Cell Junction | junction | cell junction (GO:0030054) | 68 | 16 | 34 | 50 |
| Centrosome | centrosome | centrosome (GO:0005813) | 10 | 2 | 5 | 7 |
| Cytoplasm | cytoplasm, cytoplasmic | cytoplasm (GO:0005737) | 296 | 1061 | 160 | 1197 |
| Endoplasmic Reticulum | endoplasmic, sarcoplasmic, reticulum | endoplasmic reticulum (GO:0005783) | 113 | 39 | 51 | 101 |
| Focal adhesion | focal adhesion | focal adhesion (GO:0005925) | 29 | 0 | 28 | 1 |
| Golgi Apparatus | golgi | golgi apparatus (GO:0005794) | 67 | 12 | 25 | 54 |
| Intercellular bridge | intercellular bridge | intercellular bridge (GO:0045171) | 24 | 2 | 19 | 7 |
| Intermediate filament cytoskeleton | intermediate filament cytoskeleton | intermediate filament cytoskeleton (GO:0045111) | 126 | 0 | 118 | 8 |
| Intracellular membrane-bounded organelle (vesicle[c]) | intracellular membrane-bounded organelle | Intracellular membrane-bounded organelle (GO:0043231) | 229 | 1116 | 118 | 1227 |
| Microtubule cytoskeleton | microtubule cytoskeleton | microtubule cytoskeleton (GO:0015630) | 112 | 55 | 68 | 109 |
| Microtubule end | microtubule end | microtubule end (GO:1990752) | 138 | 0 | 133 | 5 |
| Microtubule organizing center | microtubule organizing center | microtubule organizing center (GO:0005815) | 110 | 34 | 95 | 49 |
| Mitochondrion | mitochondria, mitochondrial, mitochondrion | mitochondrion (GO:0005739) | 134 | 2 | 44 | 92 |
| Nuclear membrane | nuclear membrane | nuclear membrane (GO:0031965) | 1151 | 0 | 1139 | 12 |
| Nucleolus | nucleolus, nucleolar | nucleolus (GO:0005730) | 25 | 11 | 12 | 24 |
| Nucleoplasm | nucleoplasm | nucleoplasm (GO:0005654) | 10 | 125 | 4 | 131 |
| Nucleus | nucleus, nuclei, nuclear | nucleus (GO:0005634) | 288 | 340 | 118 | 510 |
| Plasma Membrane | plasma | plasma membrane (GO:0005886) | 308 | 302 | 164 | 446 |
| | Expected representative | | | | | |
| | Unexpected representative | | | | | |

[a]Seeded nodes removed due to subgraph omission.

[b]Because subgraph nodes may root to more than one representative root node, the totals in this table do not add up to the total number of GO terms in Cellular Component.

[c]HPA conservatively annotates "vesicles" as intracellular membrane-bounded organelle.

sets that are needed. Based on these results, GOcats should offer appreciable computational improvement on significantly larger datasets. This is demonstrated in GOcats's application in annotation enrichment analysis involving all three GO sub-ontologies, which executes in just a few seconds [23].

One key feature of GOcats is the ability to easily customize category subgraphs of interest. To improve agreement and rectify potential differences in term granularity, we used GOcats to

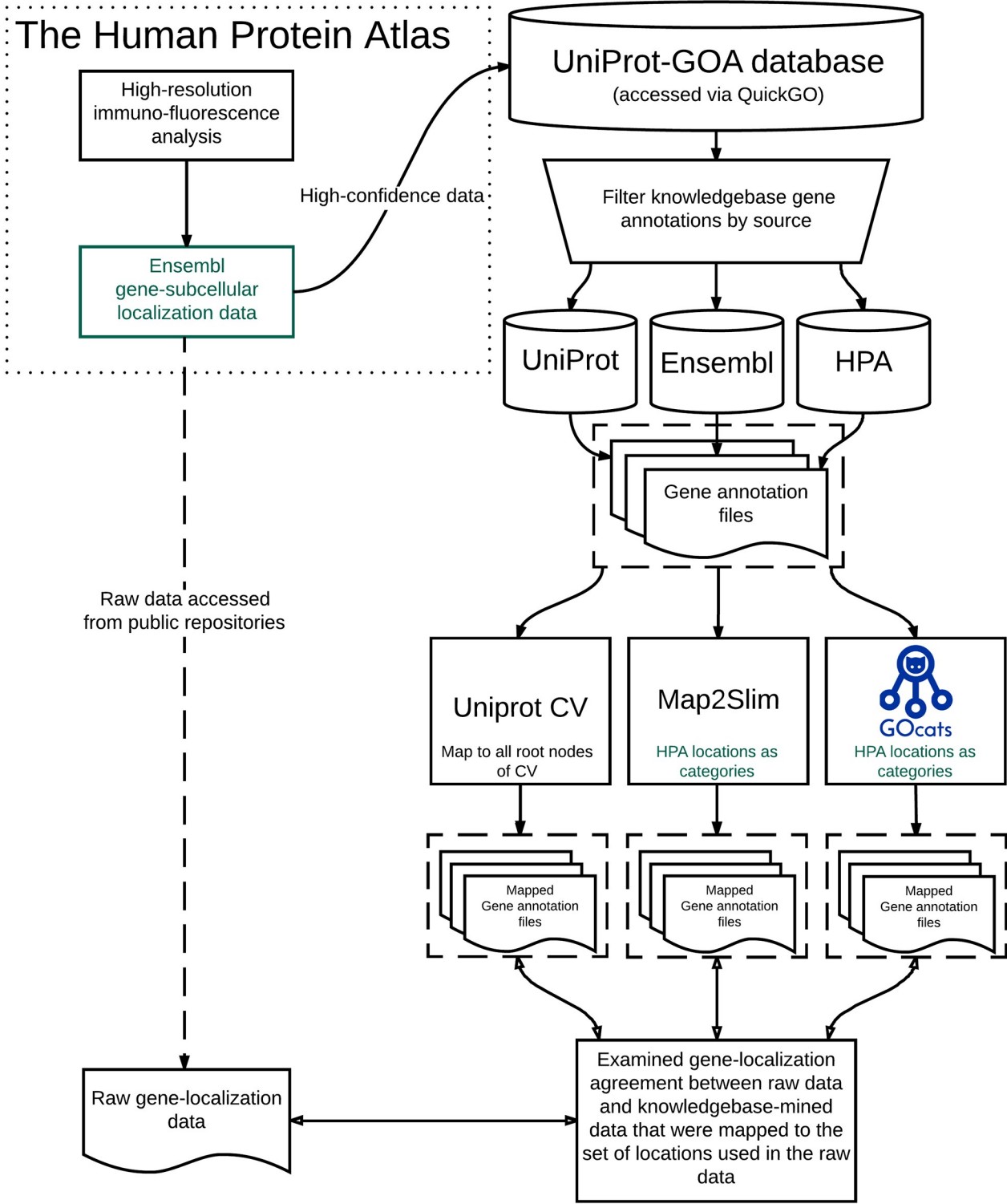

**Fig 4. Methods overview of knowledgebase gene annotation mapping and comparison to human protein database subcellular localization raw data.**

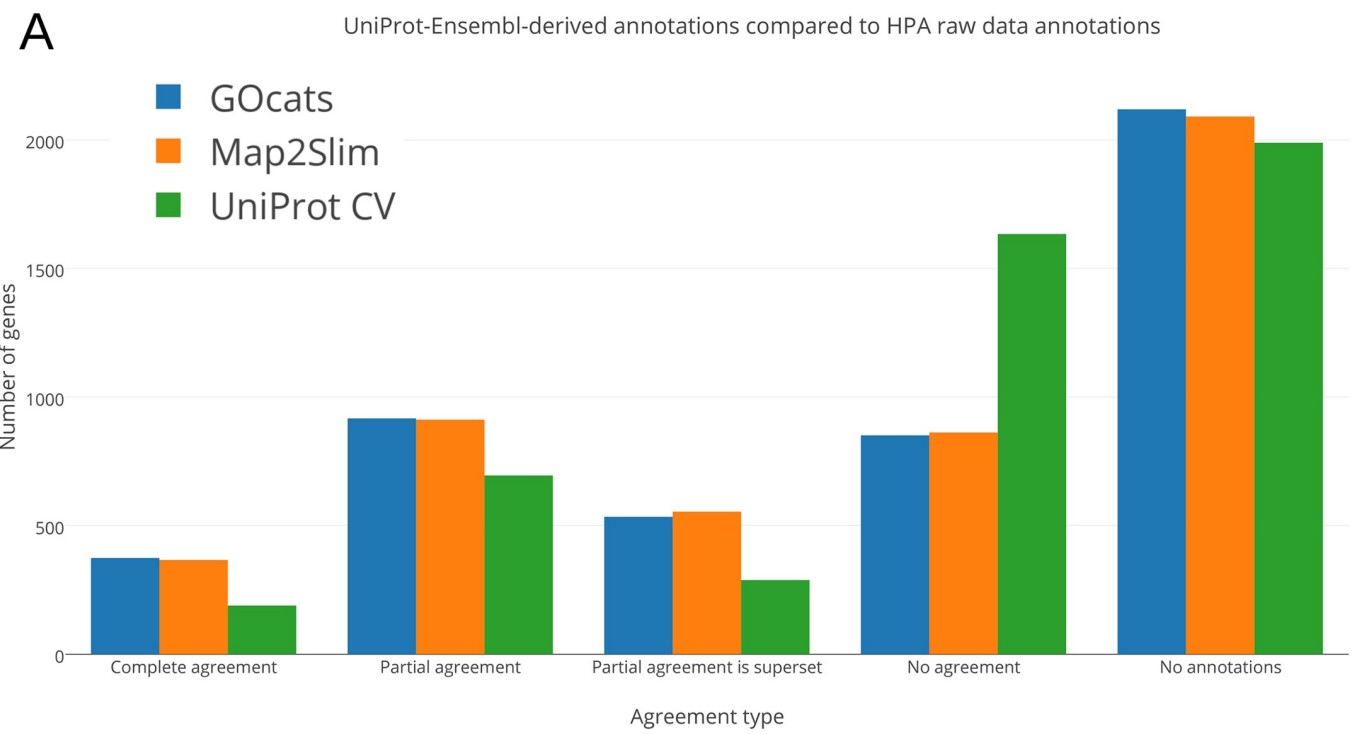

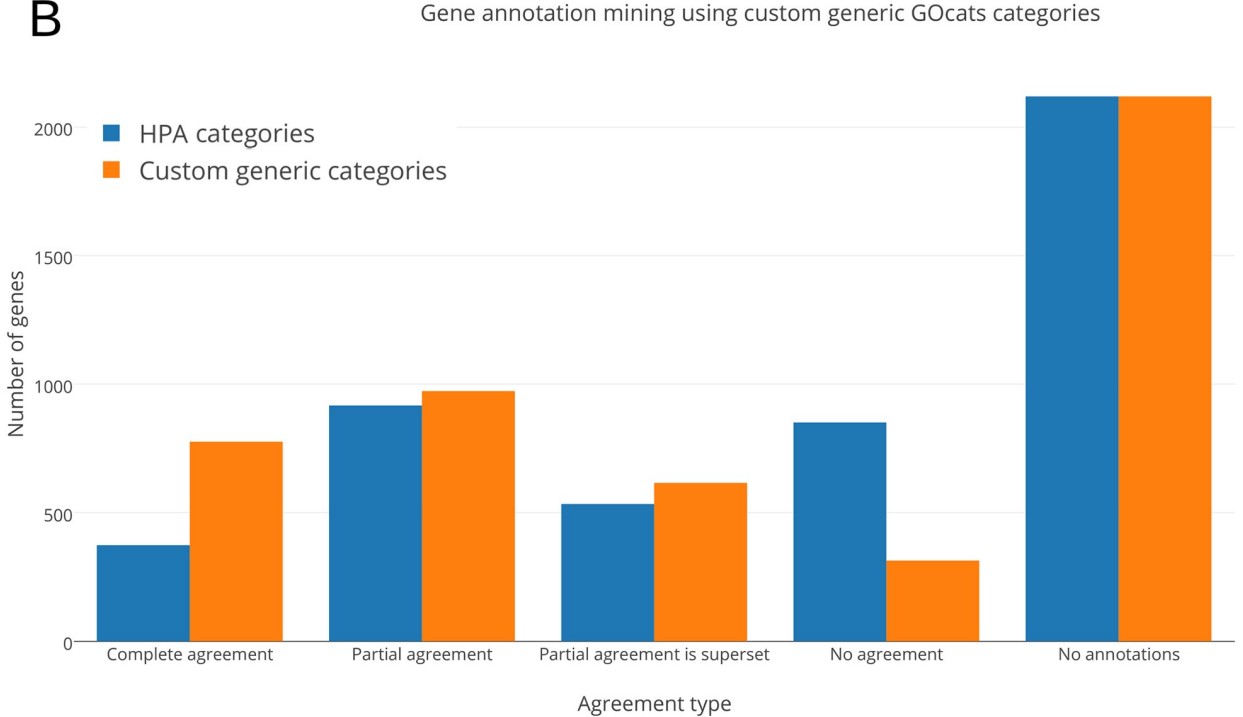

**Fig 5. Comparison of UniProt-Ensembl knowledgebase annotation data mining extraction performance by GOcats, Map2Slim, and UniProt CV.**
"Complete agreement" refers to genes where all subcellular locations derived from the knowledgebase and the HPA dataset matched, "partial agreement" refers to genes with at least one matching subcellular location, "partial agreement is superset" refers to genes where knowledgebase subcellular locations are a superset of the HPA dataset (these are mutually exclusive to the "partial agreement" category), "no agreement" refers to genes with no subcellular locations in common, and "no annotations" refers to genes in the experimental dataset that were not found in the knowledgebase. The more-generic categories used in panel B can be found in Table 3. A) Number of genes of the given agreement type when comparing mapped gene product annotations assigned by UniProt and Ensembl in the

EMBL-EBI knowledgebase to those taken from The Human Protein Atlas' raw data. Knowledgebase annotations were mapped by GOcats, Map2Slim, and the UniProt CV to the set of GO annotations used by the HPA in their experimental data. B) Shift in agreement following GOcats' mapping of the same knowledgebase gene annotations and the set of annotations used in the raw experimental data using a more-generic set of location terms meant to rectify potential discrepancies in annotation granularity.

organize HPA's raw data annotation along with the knowledgebase data into slightly more generic categories (Table 5).

In doing so, GOcats can query over twice as many knowledgebase-derived gene annotations with complete agreement with the more-generic HPA annotations, while also increasing the number of genes in the categories of "partial" and "partial agreement is superset" agreement types and decreasing the number of genes in the "no agreement" category (Fig 5B).

We then compared the methods' mapping of knowledgebase gene annotations derived from HPA to the HPA experimental dataset to demonstrate how researchers could use the GOcats suite to evaluate how well their own experimental data is represented in public knowledgebases. Because the set of gene annotations used in the HPA experimental dataset and in the HPA-derived knowledgebase annotations are identical, no term mapping occurred during the agreement evaluation and so the assignment agreement was identical between GOcats and M2S. As expected, the complete agreement category was high, although there was a surprising number of partial agreement and even some genes that had no annotations in agreement (Fig 5). We next broke down which locations were involved in each agreement type and noted that the "nucleus," "nucleolus," and "nucleoplasm" had the highest disagreement relative to their sizes, but these disagreements were present across nearly all categories (Table 6).

Both M2S and GOcats avoid superset category term mapping; neither map a category-representative GO term to another category-representative GO term if one supersedes another (although GOcats has the option to enable this functionality). Therefore, discrepancies in

**Table 5. Generic location categories used to resolve potential scoping inconsistencies in HPA raw data.**

| HPA annotation category | GOcats-customized general HPA category |
|---|---|
| Actin cytoskeleton | Cytoskeleton |
| Centrosome | |
| Intermediate filament cytoskeleton | |
| Microtubule cytoskeleton | |
| Microtubule end | |
| Microtubule organizing center | |
| Aggresome | Aggresome |
| Cell junction | Cell junction |
| Cytoplasm | Cytoplasm |
| Endoplasmic reticulum | Endoplasmic reticulum |
| Focal adhesion | Focal adhesion |
| Golgi apparatus | Golgi apparatus |
| Intercellular bridge | intercellular bridge |
| intracellular membrane-bounded organelle | intracellular membrane-bounded organelle |
| Mitochondrion | Mitochondrion |
| Nucleus | Nucleus |
| Nucleoplasm | |
| Nuclear membrane | |
| Nucleolus | Nucleolus |
| Plasma membrane | Plasma membrane |

**Table 6. Summary of gene location category agreement between manually-curated HPA raw data and GOcats/Map2Slim categorized HPA-derived annotations.**

| Location | Agreement* | | | | |
|---|---|---|---|---|---|
| | Complete | Partial | Superset‡ | None | Not in Knowledgebase |
| Actin cytoskeleton | 51 | 0 | 7 | 0 | 37 |
| Aggresome | 2 | 0 | 0 | 3 | 4 |
| Cell Junction | 36 | 0 | 17 | 0 | 51 |
| Centrosome | 58 | 3 | 17 | 0 | 49 |
| Cytoplasm | 1037 | 55 | 162 | 5 | 643 |
| Endoplasmic Reticulum | 66 | 1 | 7 | 0 | 39 |
| Focal adhesion | 27 | 5 | 9 | 0 | 17 |
| Golgi Apparatus | 159 | 5 | 43 | 0 | 137 |
| Intercellular bridge | 14 | 0 | 4 | 0 | 19 |
| Intermediate filament cytoskeleton | 18 | 1 | 4 | 0 | 23 |
| Intracellular membrane-bounded organelle | 283 | 6 | 50 | 1 | 212 |
| Microtubule cytoskeleton | 35 | 2 | 9 | 0 | 27 |
| Microtubule end | 2 | 0 | 0 | 0 | 0 |
| Microtubule organizing center | 32 | 0 | 5 | 0 | 14 |
| Mitochondrion | 263 | 4 | 55 | 0 | 154 |
| Nuclear membrane | 47 | 6 | 17 | 0 | 39 |
| Nucleolus | 266 | 10 | 69 | 6 | 163 |
| Nucleoplasm | 989 | 26 | 230 | 23 | 534 |
| Nucleus | 437 | 14 | 217 | 23 | 373 |
| Plasma Membrane | 265 | 12 | 55 | 0 | 225 |

‡Knowledgebase genes mapped to a set of categories that is a superset of those manually assigned by the HPA in raw data

*Numbers reflect how many times a location was involved in a particular agreement type; sums of all locations for an agreement category do not indicate the total number of genes for an agreement type.

annotation should not arise by term mapping methods. Nevertheless, we hypothesized that some granularity-level discrepancies exist between the HPA experimental raw data and the HPA-assigned gene annotations in the knowledgebase. We performed the same custom category generic mapping as we did for the previous test and discovered that some disagreements were indeed accounted for by granularity-level discrepancies, as seen in the decrease in "partial" and "no agreement" categories and increase in "complete" agreement category following generic mapping (Fig 6, blue bars). For example, 26S proteasome non-ATPase regulatory subunit 3 (PSMD3) was annotated to the nucleus (GO:0005634) and cytoplasm (GO:0005737) in the experimental data but was annotated to the nucleoplasm (GO:0005654) and cytoplasm in the knowledgebase. By matching the common ancestor mapping term "nucleus", GOcats can group the two annotations in the same category. In total, 132 terms were a result of semantic scoping discrepancies. Worth noting is the fact that categories could be grouped to common categories to further improve agreement, for example "nucleolus" within "nucleus."

Interestingly, among the remaining disagreeing assignments were some with fundamentally different annotations. Many of these are cases in which either the experimental data, or knowledgebase data have one or more additional locations distinct from the other. For example, NADH dehydrogenase [ubiquinone] 1 beta subcomplex subunit 6 (NDUB6) was localized only to the mitochondria (GO:0005739) in the experimental data yet has annotations to the mitochondria and the nucleoplasm (GO:0005654) in the knowledgebase. Why such discrepancies exist between experimental data and the knowledgebase is not clear.

HPA raw data-HPA knowledgebase data comparison using custom GOcats categories

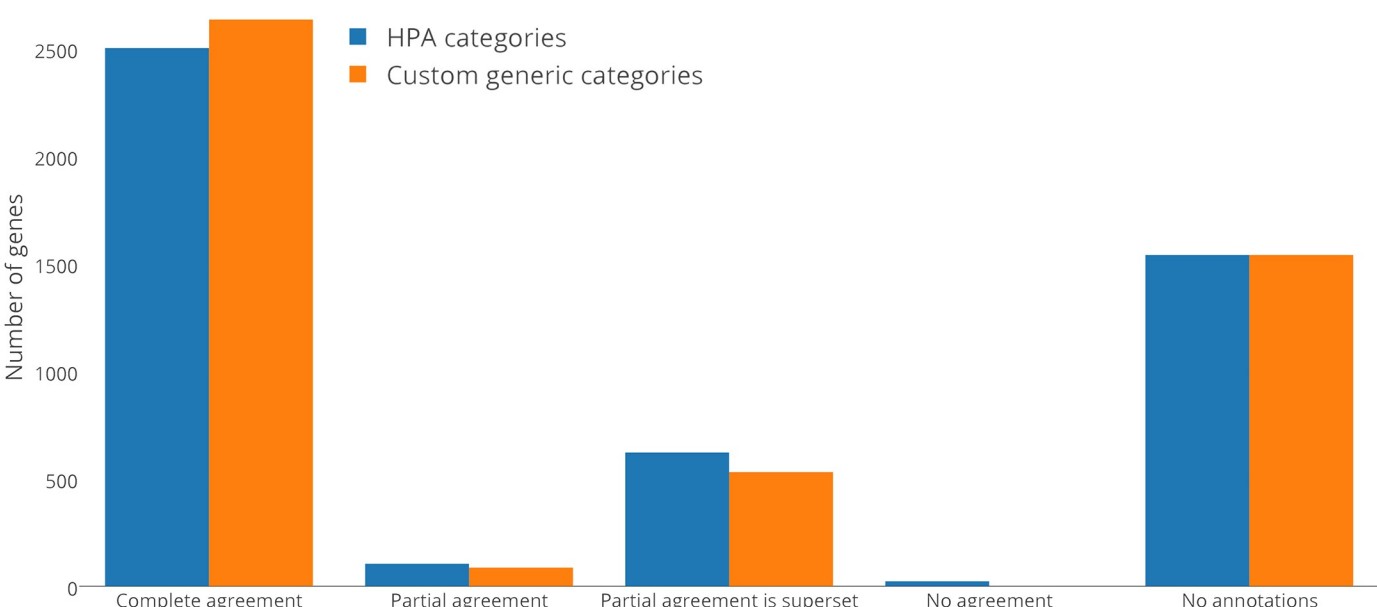

**Fig 6. Comparison of HPA knowledgebase derived annotations to HPA experimental data.** Number of genes in the given agreement type when comparing gene product annotations assigned by HPA in the EMBL-EBI knowledgebase to those in The Human Protein Atlas' raw experimental data. "Complete agreement" refers to genes where all subcellular locations derived from the knowledgebase and the HPA dataset matched, "partial agreement" refers to genes with at least one matching subcellular location, "partial agreement is superset" refers to genes where knowledgebase subcellular locations are a superset of the HPA dataset (these are mutually exclusive to the "partial agreement" category), "no agreement" refers to genes with no subcellular locations in common, and "no annotations" refers to genes in the experimental dataset that were not found in the knowledgebase. The more-generic categories used in panel B can be found in Table 3.

We were also surprised by the high number of genes with "supportive" annotations in the HPA raw data that were not found in the EMBL-EBI knowledgebase when filtered to those annotated by HPA. As Fig 6 shows, roughly one-third of the annotations from the raw data were missing altogether from the knowledgebase; the gene was not present in the knowledgebase whatsoever. This was surprising because "supportive" was the highest confidence score for subcellular localization annotation.

## Discussion

Discrepancies in the semantic granularity of gene annotations in knowledgebases represent a significant hurdle to overcome for researchers interested in mining genes based on a set of annotations used in experimental data. To demonstrate the potential GOcats has in resolving these discrepancies, we categorized annotations from HPA-sourced gene annotations using GOcats, M2S, and the UniProt subcellular localization CV. The HPA source was chosen because primary data from high-throughput immunofluorescence-based gene product localization experiments exist in publicly-accessible repositories and have been inspected by experts and given a confidence score [10]. As we show, utilizing only the set of specific annotations used in the HPA's experimental data, M2S's mapping matches only 366 identical sets of gene annotations from the knowledgebase with GOcats matching slightly more (Fig 5A). GOcats alleviates this problem by allowing researchers to define categories at a custom level of granularity so that categories may be specific enough to retain biological significance, but generic enough to encapsulate a larger set of knowledgebase-derived annotations. When we reevaluated the agreement between the

raw data and knowledgebase annotations using custom GOcats categories for "cytoskeleton" and "nucleus", the number of identical gene annotations increased to 776 (Fig 5B).

Because GOcats relies on user-input keywords to define categories, we understand that there is a risk of adding user bias when applying this method to organizing results of various analyses. While we have taken care to avoid bias in the comparisons made in this report, for example citing the exact category defining GO term for each category compared between methods (Fig 3, Tables 2 and 3) and reporting the exact common-sense categorizations applied when grouping location categories from HPA (Table 5), we strongly caution users to exercise similar care in their use as well. For instance, when categorizing results from annotation enrichment analyses it may be tempting to filter results to those categories defined by the user, which might conveniently eliminate unexpected (unwanted) highly-enriched terms. We do not condone the use of GOcats in this way. But because GOcats will always produce the same subgraph categorizations for the same set of keywords used with the same version of GO, we argue that our categorization is more reproducible and less prone to bias than manually grouping GO terms into categories or otherwise manually identifying major concepts represented from omics-level analyses. Furthermore, the set of keywords can be provided along with the version of GOcats, GO, and the dataset to enable reproducibility of analyses by others.

As GO continues to grow, automated methods to evaluate the structural organization of data will become necessary for curation and quality control. Because GOcats allows versatile interpretation of the GO directed acyclic graph (DAG) structure, it has many potential curation and quality control uses, especially for evaluating the high-level ontological organization of GO terms. For example, GOcats can facilitate the integrity checking of annotations that are added to public repositories by streamlining the process of extracting categories of annotations from knowledgebases and comparing them to the original annotations in the raw data. Interestingly, about one-third of the genes annotated with high-confidence in the HPA raw data were missing altogether from the EMBL-EBI knowledgebase when filtered to the HPA-sourced annotations. While this surprised us, the reason appears to be due to HPA's use of two separate criteria for "supportive" annotation reliability scores and for knowledge-based annotations. For "supportive" reliability, one of several conditions must be met: i) two independent antibodies yielding similar or partly similar staining patterns, ii) two independent antibodies yielding dissimilar staining patterns, both supported by experimental gene/protein characterization data, iii) one antibody yielding a staining pattern supported by experimental gene/protein characterization data, iv) one antibody yielding a staining pattern with no available experimental gene/protein characterization data, but supported by other assay within the HPA, and v) one or more independent antibodies yielding staining patterns not consistent with experimental gene/protein characterization data, but supported by siRNA assay [10]. Meanwhile knowledge-based annotations are dependent on the number of cell lines annotated; specifically, the documentation states, "Knowledge-based annotation of subcellular location aims to provide an interpretation of the subcellular localization of a specific protein in at least three human cell lines. The conflation of immunofluorescence data from two or more antibody sources directed towards the same protein and a review of available protein/gene characterization data, allows for a knowledge-based interpretation of the subcellular location" (Uhlen et al., 2015). Unfortunately, we were unable to explore these differences further, since the experimental data-based subcellular localization annotations appeared aggregated across multiple cell lines, without specifying which cell lines were positive for each location. Meanwhile, tissue- and cell-line specific data, which contained expression level information, did not also contain subcellular localizations. Therefore, we would suggest that HPA and other major experimental data repositories always provide a specific annotation reliability category in their distilled experimental datasets that matches the criteria used for deposition of derived annotations in the

knowledgebases. Such information will be invaluable for performing knowledgebase-level evaluation of large curated sets of annotations. One step better would involve providing a complete experimental and support data audit trail for each derived annotation curated for a knowledgebase, but this may be prohibitively difficult and time-consuming to do.

Looking towards the future, the work demonstrated here is a critical first step towards a goal of automatically enumerating all representable concepts within GO. Such an enumeration would provide scientists with the usable set of GO-representable concept subgraphs for a large variety of analyses unbiased by human selection. GOcats can derive subgraphs representing a specific concept by utilizing keywords and key terms, which would be a major component for an overall method to enumerate all representable concepts. We expect two other major components will be required, first is a way to derive possible key words and key terms and the last is a way to evaluate the quality of the concept subgraphs that are generated. We expect the latter evaluation to involve the development of various graph-based metrics for this purpose.

## Conclusions

In this study, we: i) demonstrated an improvement in retrievable ontological information content by the reevaluation of GO's has_part relation ii) applied our new method GOcats toward the categorization and utilization of the GO Cellular Component sub-ontology, and iii) evaluated the ability of GOcats and other mapping tools to relate HPA experimental to HPA knowledgebase GO Cellular Component annotation sources. GOcats outperforms the UniProt CV with respect to accurately deriving gene-product subcellular location from the UniProt and Ensembl database with the HPA raw dataset of gene localization annotations treated as the gold standard (Fig 5A). Moreover, the comparison of GOcats to M2S demonstrates similar mapping performance between the two methods, but with GOcats providing important improvements in mapping, computational speed, ease of use, and flexibility of use. In a previous publication, we demonstrated an improvement in the statistical power of gene-annotation enrichment analyses using GOcats along with all GO sub-ontologies [23].

In conclusion, GOcats enables the user to create custom, GO slim-like filters to map fine-grained gene annotations from GAFs to general subcellular compartments without needing to hand-select a set GO terms for categorization. Moreover, users can use GOcats to quickly customize the level of semantic specificity for annotation categories. Furthermore, GOcats was designed for scientists who are less familiar with GO; however, the package has advanced features for users with more bioinformatics expertise. GOcats enables a safe and more comprehensive semantic scoping utilization of go-core, preventing mistakes that can easily arise from using go-core instead of go-basic. Together, these improvements can impact a variety of GO knowledgebase data mining use-cases as well as knowledgebase curation and quality control. Looking towards the future, GOcats provides a critical categorization method for a future automatic enumeration of all representable concepts within GO.

## Methods

### Methodological overview and design rationale

We designed GOcats with a biologist user in mind, who may not be aware of the dangers associated with using different versions of GO for organizing terms with tools like M2S or how to circumvent potential pitfalls. For instance, although the M2S documentation (https://github.com/owlcollab/owltools/wiki/Map2Slim) states, "We recommend the go-basic version of the ontology be used, which contains: subClassOf (is a), part of, regulates (+ positively and negatively regulates)" and, "You can also use the full version of GO and filter those relationships you do not want to consider," a non-bioinformatician may not be aware of how to filter out

relationships from GO in a way that is safe to use the tool—or, more pertinently—the user may wish to use a fuller extent of the information contained in the ontology when organizing their terms. Currently, GOcats version 1.1.4 can handle go-core's is_a, part_of, and has_part relations, with the has_part reinterpreted to retain proper scoping semantics, as detailed below and elsewhere [23]. As the development of GOcats progresses, we plan on handling the organization of terms connected by additional relations such as negatively/positively_regulates.

GOcats uses the go-core version of the GO database, which contains relations that connect the separate ontologies and may point away from the root of the ontology. GOcats can either exclude non-scoping relations or invert has_part directionality into a part_of_some interpretation, maintaining the acyclicity of the graph. Therefore, it can represent go-core as a DAG.

GOcats is a Python package written in major version 3 of the Python program language [26] and available on GitHub and the Python Package Index. It uses a Visitor design pattern implementation [27] to parse the go-core Ontology database file [4]. Searching with user-specified sets of keywords for each category, GOcats extracts subgraphs of the GO DAG and identifies a representative node for each category in question and whose child nodes are detailed features of the components. Fig 7 illustrates this approach, and details follow in pseudocode.

To overcome issues regarding scoping ambiguity among mereological relations, we assigned properties indicating which term was broader in scope and which term was narrower in scope to each edge object created from each of the scope-relevant relations in GO. For example, in the node pair connected by a part_of or is_a edge (e.g. node 1 is_a node 2), node 1 is narrower in scope than node 2. Conversely, node 1 is broader in scope than node 2 when connected by a has_part edge (e.g. node 1 has_part node 2). This edge is therefore reinterpreted by GOcats as part_of_some. **This reinterpretation is not meant to imply exclusivity in composition between the meronym and the holonym. It simply stands as a distinction between "part of all" which is what the current "part_of" relationship implies, and "part of some," or to be more verbose "instance a is part of instance b in at least one known biological example."** We have described additional explanations and rationale for this re-interpretation elsewhere and demonstrate improvement in annotation enrichment analyses across GO Cellular Component, Molecular Function, and Biological Process sub-ontologies, when this re-interpretation is used [23].

While the default scoping relations in GOcats are is_a, part_of, and has_part, the user has the option to define the scoping relation set. For instance, one can create go-basic-like subgraphs from a go-core version ontology by limiting to only those relations contained in go-basic. For convenience, we have added a command line option, "go-basic-scoping," which allows only nodes with is_a and part_of relations to be extracted from the graph. Detailed API documentation and user-friendly tutorials are available online (https://gocats.readthedocs.io/en/latest/).

For mapping purposes, Python dictionaries are created which map GO terms to their corresponding category or categories. For inter-subgraph analysis, another Python dictionary is created which maps each category to a list of all its graph members. By default, fine-grained terms do not map to category root-nodes that define a subgraph that is a superset of a category with a root-node nearer to the term. For example, a member of the "nucleolus" subgraph would map only to "nucleolus," and not to both "nucleolus" and "nucleus". However, the user also has the option to override this functionality if desired with a simple "—map-supersets" command line option. Furthermore, we've included the option for users to directly input GO terms as category representatives, should they not wish to use keywords to define subgraph categories. This is helpful for users who have already compiled lists of GO terms by hand for use with other tools.

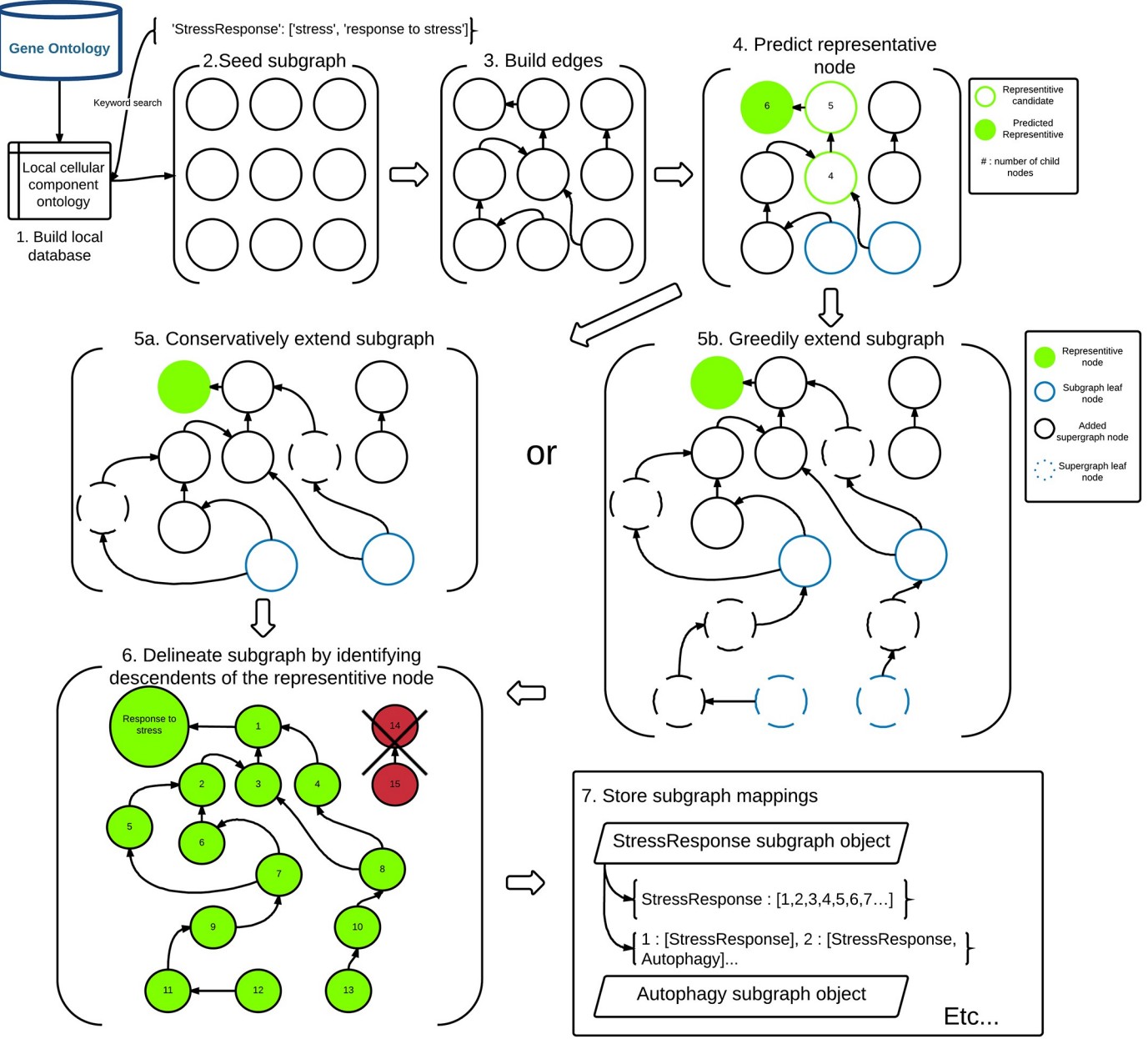

**Fig 7. Flowchart of the GOcats' subgraph creation method.**

## Implementation overview

As illustrated in the UML diagram in Fig 8A, the GOcats package is implemented using several modules that have clear dependencies starting from a command line interface (CLI) in gocats. py which depend on most of the other modules including ontologyparser.py, godag.py, sub-dag.py and tools.py. GOcats uses 10 classes implemented across ontologyparser.py, godag.py, subdag.py, and dag.py to extract and internally represent the GO database. GoParser, which inherits from the base OboParser class (Fig 8B), utilizes a visitor design pattern and regular expressions to parse the flat GO database obo file and instantiate the objects necessary to

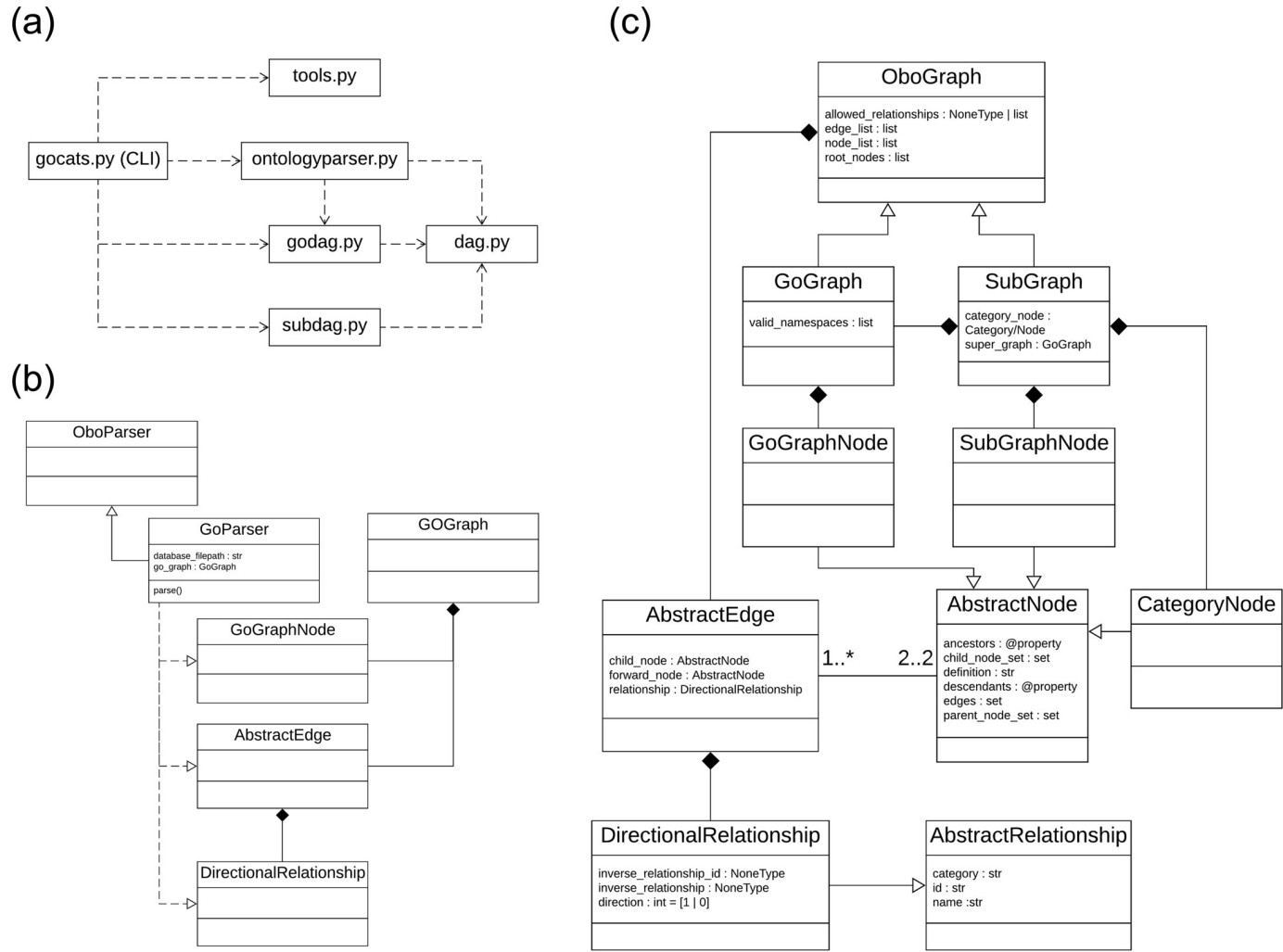

**Fig 8. UML diagrams describing the GOcats implementation.** A) UML module dependency diagram. B) UML class diagram of GO database parsing classes. C) UML class diagram of the GO graph representation.

represent the GO DAG structure. These instantiated objects include (Fig 8C): 1) the GoGraph container object for the parts of the graph, which inherits from a more generic OboGraph containing functions for adding, removing, and modifying nodes and edges; 2) GoGraphNode objects for representing each term parsed from the ontology, which inherits from Abstract-Node; 3) AbstractEdge objects for representing each instance of a relation parsed from the ontology; and 4) DirectionalRelationship objects, which inherit from the more generic AbstractRelationship object for representing each *type* of directional relation encountered in the ontology (for GO, all relations are directional, and this distinction is made only in anticipation for future extensions to handle other ontologies).

AbstractEdge objects and AbstractNode objects contain references to one another, which simplifies the process of iterating through ancestor and descendant nodes and allows for functions such as AbstractEdge.connect_nodes, which requires that the edge object update the node object's child_node_set and parent_node_set. In this context, AbstractNode is a true abstract base class, while AbstractEdge started out as an abstract base class but eventually

became a concrete class during development. However, we see the possibility of AbstractEdge becoming a base class in the future.

Ancestors and descendants of a node are implemented as sets, which are lazily created through the use of a Python property decorator (i.e. Python's preferred "getter" syntax). At the first access of these sets through the ancestor or descendent property, the set is calculated with a recursive algorithm, stored for future use, and returned for immediate access. Subsequent accesses simply return the stored set. If the set of edges within a node change, the ancestor and descendent node sets will be recalculated on their next access. This implementation prevents pre-calculation of these sets when they are not used, while enabling their reuse within efficient graph analysis methods.

AbstractEdge also contains a reference to a DirectionalRelationship object, which is critical for graph traversal. This is because DirectionalRelationship contains the true directionality of the mereological correspondence between the categorization relevant relations (is_a, part_of, and has_part). In other words, it is within this class that we define in which direction the edge should be traversed when categorizing terms. Currently these rules are hard-coded within GoParser's relationship_mapping dictionary.

The gocats.py module (Fig 8A) implements the command line interface and is responsible for handling the command line arguments, using the provided keywords and specified arguments like namespace filters (e.g. Cellular Component, Molecular Function, and Biological Process) to instantiate a GoParser object, a GoGraph object and a SubGraph object for each set of provided keywords. After creation of the GoGraph internal representation, each category subgraph is created by first instantiating the SubGraph object and calling the from_filtered_-graph function, which filters to those nodes from the GoGraph containing the keywords in their names and definition. Note that the SubGraph object and GoGraph object both inherit from OboGraph, and that the SubGraph object contains a reference to GoGraph object (super-graph data member) of which it is a subgraph. This design was implemented to avoid accidental alterations of the GoGraph object when altering the contents of the subgraph, and to allow for specialization of functions within SubGraph without needing to use unique names e.g. add_node(). GoGraphNode objects within the subgraph are wrapped by SubGraphNode objects, which are directly used by the SubGraph object, but retain all original properties such as name, definition, and sets of edge object references, otherwise insidious changes could occur to the GoGraph object when updating the SubGraph object. The SubGraph object also contains a CategoryNode object, which wraps the category representative GoGraphNode object(s) for the subgraph category.

## Specific implementation details

User-provided keyword sets are used by GOcats to query GO terms' name and definition fields to create an initial seeding of the subgraph with terms that contain at least one keyword. This seeding is a list of nodes from the whole go-core graph (supergraph) that pass the query. Node synonyms were not used, due to there being four types of synonyms in GO: exact, narrow, broad, and related. Also, many nodes within GO do not have synonyms, which may create an unequal utilization of nodes if synonyms were queried. However, in the future, synonym utilization for seeding purposes may be revisited.

```
FOR node in supergraph.nodes
  IF keyword from keyword_list in node.name or node.definition
    subgraph.seeding_list.append(node)
```

Using the graph structure of GO, edges between these seed nodes are faithfully recreated except where edges link to a node that does not exist in the set of newly seeded GO terms.

During this process, edges of appropriate scoping relations are used to create children and parent node sets for each node.

```
FOR edge in supergraph.edges
  IF edge.parent_node in subgraph.nodes AND edge.child_node
in subgraph.nodes AND /
    edge.relation is TYPE: SCOPING
    subgraph.edges.append(edge)
  ELSE
    PASS
FOR subnode in subgraph.nodes
  subnode.child_node_set = /
    {child_node for child_node in supergraph.id_index[sub-
node.id].child_node_set /
    if child_node.id in subgraph.id_index}
  subnode.parent_node_set = /
    {parent_node for parent_node in supergraph.id_index[sub-
node.id].parent_node_set /
    if parent_node.id in subgraph.id_index}
```

GOcats then selects a category representative node to represent the subgraph. To do this, a list of candidate representative nodes is compiled from non-leaf nodes, i.e. root-nodes in the subgraph which have at least one keyword in the term name. A single category representative root-node is selected by recursively counting the number of children each candidate term has (i.e. creating the node.descendents) and choosing the term with the most children.

```
FOR subnode in subgraph.nodes
  IF subnode.child_node_set ! = None AND ANY keyword in sub-
node.name
    candidate_list.append(subnode)
  ELSE
    PASS
representative_node = MAX(LEN(node.descendants) FOR node in
candidates)
```

Because it may be possible that highly-specific or uncommon features included in the GO may not contain a keyword in its name or definition but still may be part of the subgraph in question by the GO graph structure, GOcats re-traces the supergraph to find various node paths that reach the representative node. We have implemented two methods for this subgraph extension: i) comprehensive (greedy) extension, whereby all supergraph descendants of the representative node are added to the subgraph and ii) conservative extension, whereby the supergraph is checked for intermediate nodes between subgraph leaf nodes and the subgraph representative node that may not have seeded in the initial step.

```
Comprehensive (Greedy) extension:
FOR node in supergraph.nodes
  IF ANY (ancestor_node in node.ancestors) in subgraph
    subgraph.nodes.append(ancestor_node)
UPDATE subgraph # appropriate edges added and parent/child
nodes assigned
Conservative extension:
FOR leaf_node in subgraph.leaf_nodes # nodes with no children
  start_node = leaf_node
  end_node = representative_node
```

```
    FOR node in super_graph.start_node.ancestors ∩ supergraph.
end_node.descendents
        subgraph.nodes.append(node)
    UPDATE subgraph # appropriate edges added and parent/child
nodes assigned
```

The subgraph is finally constrained to the descendants of the representative node in the subgraph. This excludes unrelated terms that were seeded by the keyword search due to serendipitous keyword matching.

## Creating category mappings from UniProt's subcellular location controlled vocabulary

We created mappings from fine-grained to general locations in UniProt's subcellular location CV [2] for comparison to GOcats. To accomplish this, we parsed and recreated the graph structure of UniProt's subcellular locations CV file [13] in a manner similar to the parsing of GO (Fig 2). Briefly, the flat-file representation of the CV file is parsed line-by-line and each term is stored in a dictionary along with information about its graph neighbors as well as its cross-referenced GO identifier. We assumed that terms without parent nodes in this graph are category-defining root-nodes and created a dictionary where a root-node key links to a list of all recursive children of that node in the graph. Only those terms with cross-referenced GO identifiers were included in the final mapping. The category subgraphs created from UniProt were compared to those with corresponding category root-nodes made by GOcats. An inclusion index, $I$, was calculated by considering the two subgraphs' members as sets and applying the following equation:

$$I = \frac{|S_n \cap S_g|}{|S_n|} \tag{1}$$

where $S_n$ and $S_g$ are the set of members within the non-GOcats-derived category and GOcats-derived category, respectively. It is worth noting here that the size of the UniProt set was always smaller than the GOcats set. This is due to the inherent size differences between UniProt's CV and the Cellular Component sub-ontology.

## Creating category mappings from Map2Slim

The Java implementation of OWLTools' M2S does not include the ability to output a mapping file between fine-grained GO terms and their GO slim mapping target from the GAF that is mapped. To compare subgraph contents of GOcats categories to a comparable M2S "category," we created a special custom GAF where the gene ID column and GO term annotation column of each line were each replaced by a different GO term for each GO term in Cellular Component, data-version: releases/2016-01-12. We then allowed M2S to map this GAF with a provided GO slim. The resulting mapped GAF was parsed to create a standalone mapping between the terms from the GO slim and a set of the terms in their subgraphs.

## Mapping gene annotations to user-defined categories

To allow users to easily map gene annotations from fine-grained annotations to specified categories, we added functionality for accepting GAFs as input, mapping annotations within the GAF and outputting a mapped GAF into a user-specified results directory. The input-output scheme used by GOcats and M2S are similar, with the exception that GOcats accepts the mapping dictionary created from category keywords, as described previously, instead of a GO slim.

GAFs are parsed as a tab-separated-value file. When a row contains a GO annotation in the mapping dictionary, the row is rewritten to replace the original fine-grained GO term with the corresponding category-defining GO term. If the gene annotation is not in the mapping dictionary, the row is not copied to the mapped GAF, and is added to a separate file containing a list of unmapped genes for review. The mapped GAF and list of unmapped genes are then saved to the user-specified results directory.

### Visualizing and characterizing intersections of category subgraphs

To compare the contents of category subgraphs made by GOcats, UniProt CV, and M2S, we took the set of subgraph terms for each category in each method, converted them into a Pandas DataFrame [28] representation, and plotted the intersections using the UpSetR R package [25]. Inclusion indices were also computed for M2S categories using Eq 1. Jaccard indices were computed for every subgraph pair to evaluate the similarity between subgraphs of the same concept, created by different methods.

### Assigning generalized subcellular locations to genes from the knowledgebase and comparing assignments to experimentally-determined locations

We first mapped two GAFs downloaded from the EMBL-EBI QuickGO resource [12] using GOcats, the UniProt CV, and M2S. We filtered the gene annotations by dataset source and evidence type, resulting in separate GAFs containing annotations from the following sources: UniProt-Ensembl, and HPA. Both GAFs had the evidence type, inferred from Electronic Annotation, filtered out because it is generally considered to be the least reliable evidence type for gene annotation and in the interest of minimizing memory usage. We used this data to assess the performance of the mapping methods in their ability to assign genes to subcellular locations based on annotations from knowledgebases by comparing these assignments to those made experimentally in HPA's localization dataset (Fig 3A). Comparison results for each gene were aggregated into 4 types: i) "complete agreement" for genes where all subcellular locations derived from the knowledgebase and the HPA dataset matched, ii) "partial agreement" for genes with at least one matching subcellular location, iii) "partial superset" for genes where knowledgebase subcellular locations are a superset of the HPA dataset, iv) "no agreement" for genes with no subcellular locations in common, and v) "no annotations" for genes in the experimental dataset that were not found in the knowledgebase.

Only gene product localizations from the HPA dataset with a "supportive" confidence score were used for this analysis (n = 4795). We created a GO slim by looking up the corresponding GO term for each location in this dataset with the aid of QuickGO term basket and filtering tools. The resulting GO slim served as input for the creation of mapped GAFs using M2S. To create mapped GAFs using GOcats, we entered keywords related to each location in the HPA dataset (Table 4). We matched the identifier in the "gene name" column of the experimental data with the identifier in the "database object symbol" column in the GAF to compare gene annotations. Our assessment of comparing the HPA raw data to mapped gene annotations from the knowledgebase represents the ability to accurately query and mine genes and their annotations from the knowledgebase into categories of biological significance. Our assessment of comparing the methods' mapping output to the HPA raw dataset represents the ability of these methods to evaluate the representation of HPA's latest experimental data as it exists in public repositories.

### Running time tests

For comparing the runtimes of GOcats and M2S for categorizing HPA's subcellular location dataset, each method was run separately on the same machine with the following

configuration: Intel ® Core ™ i7-4930K CPU with 6 hyperthreaded cores clocked at 3.40GHzn and 64 GB of RAM clocked at 1866 MHz. We used the Linux "time" command with no additional options and reported the *real* time from its output. The datasets and scripts used can be found in our FigShare (See Availability of Data and Material). We used the dataset contained in our ScriptsDirectory/KBData/11-02-2016/hpa-no_IEA.goa for these comparisons. For M2S we executed a custom script that can be found within ScriptsDirectory/runscripts:

```
sh owlmultitest.sh
```
which ran the following command, found in the same subdirectory, 50 times:
```
time sh owltoolsspeedtest.sh
```
For GOcats, we executed a custom script that can be found within ScriptsDirectory/runscripts:
```
sh gcmultitest.sh
```
which ran the following command, found in the same subdirectory, 50 times:
```
time sh GOcatsspeedtest.sh
```
Both tests were executed using the same version of the go-core used across all other analyses performed in this work, which is data version: releases/2016-01-12.

## Supporting information

**S1 Data. Visualizing the degree of overlap between the category subgraphs created by GOcats, Map2Slim, and the UniProt CV (additional categories).**
(DOCX)

**S2 Data. List of GO terms mapped by Map2Slim to the term plasma membrane that were not mapped to this location by GOcats.**
(DOCX)

**S1 File.**
(DOCX)

## Acknowledgments

We thank Dr. Robert M. Flight for his advice and expertise regarding the statistics reported in this project, for the generation of the plots in Fig 3, and for his feedback during the drafting of the manuscript. We thank Dr. Thilakam Murali for extensive feedback on the general scientific readability of the manuscript.

## Author Contributions

**Conceptualization:** Hunter N. B. Moseley.

**Data curation:** Eugene W. Hinderer, III.

**Formal analysis:** Eugene W. Hinderer, III.

**Investigation:** Eugene W. Hinderer, III.

**Project administration:** Hunter N. B. Moseley.

**Supervision:** Hunter N. B. Moseley.

**Validation:** Eugene W. Hinderer, III, Hunter N. B. Moseley.

**Writing – original draft:** Eugene W. Hinderer, III.

**Writing – review & editing:** Hunter N. B. Moseley.

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
