## [Decision Letter · Decision Letter 0]

28 Feb 2020

PONE-D-20-00699

GOcats: A tool for categorizing Gene Ontology into subgraphs of user-defined concepts

PLOS ONE

Dear Dr. Moseley,

Thank you for submitting your manuscript to PLOS ONE. After careful consideration, we feel that it has merit but does not fully meet PLOS ONE’s publication criteria as it currently stands. Therefore, we invite you to submit a revised version of the manuscript that addresses the points raised during the review process.

As you will see, both reviewers appreciated your work and considered it a welcome addition to the literature, but had important remarks concerning the relation of your work to previous work, its impact on downstream analyses, and the presentation of the manuscript. All of these remarks are constructive and helpful, and I invite you to take them into account in your revision.

We would appreciate receiving your revised manuscript by Apr 13 2020 11:59PM. To enhance the reproducibility of your results, we recommend that if applicable you deposit your laboratory protocols in protocols.io, where a protocol can be assigned its own identifier (DOI) such that it can be cited independently in the future. For instructions see: http://journals.plos.org/plosone/s/submission-guidelines#loc-laboratory-protocols

We look forward to receiving your revised manuscript.

Kind regards,

Marc Robinson-Rechavi

Academic Editor

PLOS ONE

Journal Requirements:

1. Thank you for including your funding statement; "This work was supported in part by grants NSF 1419282 (Moseley), NIH 1U24DK097215-01A1 (Higashi, Fan, Lane, Moseley), and NIH UL1TR001998-01 (Kern)."

2. 

We note that your BSD-3 License of the software may have copyright restrictions. All PLOS content is published under the Creative Commons Attribution License (CC BY 4.0), which means that the manuscript, images, and Supporting Information files will be freely available online, and any third party is permitted to access, download, copy, distribute, and use these materials in any way, even commercially, with proper attribution. For more information, see our copyright guidelines: http://journals.plos.org/plosone/s/licenses-and-copyright.

We require you to either (1) present written permission from the copyright holder to publish these figures or software specifically under the CC BY 4.0 license, or (2) remove the figures from your submission:

1.    You may seek permission from the original copyright holder of the software to publish the content specifically under the CC BY 4.0 license.

Reviewers' comments:

Reviewer's Responses to Questions

**Comments to the Author**

1. Is the manuscript technically sound, and do the data support the conclusions?

Reviewer #1: Yes

Reviewer #2: Partly

2. Has the statistical analysis been performed appropriately and rigorously? 

Reviewer #1: N/A

Reviewer #2: N/A

3. Have the authors made all data underlying the findings in their manuscript fully available?

Reviewer #1: Yes

Reviewer #2: Yes

4. Is the manuscript presented in an intelligible fashion and written in standard English?

Reviewer #1: Yes

Reviewer #2: Yes

5. Review Comments to the Author

Reviewer #1: The authors present GOcats, a tool that categorizes the Gene Ontology (GO) into subgraphs based on user inputs. Similar to GO-slim, it generates a sub-graph of the full GO using a list of user-provided keywords, while also handling the semantic scoping of the relationships within the GO. The paper itself only focuses on the sub-ontology under subcellular locations in order to be able report its accuracy against immunohistochemistry datasets in the Human Protein Atlas (HPA). Overall, it aims to reduce the manual effort required in hand-selecting a set of GO terms for categorization, which can improve the current complex workflows of GO analysis.

The paper is well-written, with an appreciable level of background and sufficient details for a scientist who is not completely familiar with the GO. The authors present how their work fits in among the numerous GO tools out there, and provide several examples of how the output of GOcats improve the correspondence between the annotations sources. The source code, scripts, and documentation are complete and appear up-to-date with the manuscript. The Methods section has included important details about their implementation, accompanied with reasonable explanation. For example, they have considered different user scenarios such as comprehensive and conservative subgraph extensions. As a statistician, I appreciate their discussion regarding how user-input keywords that prune the GO can increase the bias during enrichment analysis. A well-written software developed with the intent to maintain statistical reproducibility can also more generally benefit the field of data analysis. Overall, I would recommend this paper for publication, conditioned on the authors addressing some of my concerns below and some minor suggestions that could improve readability.

Major concerns:

1. Distinction between this manuscript and their previous one [Hinderer EW, Flight RM, Dubey R, Macleod JN, Moseley HNB. Advances in gene ontology utilization improve statistical power of annotation enrichment. PLoS One. 2019;14(8):1–20.]

a. From what I understood, the previous manuscript also introduces GOcats as a tool that ‘organizes the Gene Ontology into subgraphs representing user-defined concepts, while ensuring that all appropriate relations are congruent with respect to scoping semantics.’ Is the underlying algorithm improved in any way? Was the previous software unable to perform the same analyses done in this paper?

b. At the end of the introduction, they mention ‘In a prior publication, we demonstrated GOcats’s ability to improve gene-annotation enrichment analyses, involving all GO sub-ontologies (23).’ I suggest that either (i) they should reference (23) earlier in the introduction to make it clear when it was initially developed, or (ii) they could discuss the differences between results from the previous paper and those in this paper. I skimmed the previous paper and indeed did not find any overlap, but I think it is the authors’ responsibly to clarify the differences to their previous work in this follow-up manuscript.

2. Technical correctness of the algorithms. The pseudo-code in the Methods are interspersed and difficult to read without cross referencing. Here are some detailed pointers for the authors:

a. They mention ‘A single category representative root-node is selected by *recursively* counting the number of children each candidate term has and choosing the term with most children’. It is technically wrong to say the code block following the description recursive, because it iterative and does not consider the children of newly appended nodes.

b. The object class definitions are inconsistent. I see both ‘FOR subnode in subgraph.nodes’ vs. ‘FOR subnode in subgraph’, and similar inconsistencies for the supergraph class in different code blocks. Although I couldn’t find their corresponding source code for these steps, I strongly suggest the authors to double-check their source and make the pseudo-code reflect their implementation more precisely.

c. There are very long one-line codes such as ‘subnode.parent_node_set = …’ which are written in Python3 set-comprehension form. This is unnecessary for pseudo-code and makes it quite difficult parse the logic.

3. Time complexity and scalability.

a.When comparing themselves with M2S, they say even though they were implemented in Python and M2S was implemented in Java, they were faster in performance due to ‘the utilization of stored ancestor and descendent node’. The statement is not convincing because M2S also stores the data but perhaps in a different format. They should be clear, for example, if it is because the node information is faster to access because they are in memory when the subgraph is built, or is it because they have a better way to parse the flat files that both M2S and they use. Right now, this discussion sounds too vague and mysterious to me. So instead of letting a reader guess what is happening, they should add just a couple more reasons accounting for any overhead that they and M2S do.

b.They say that ‘GOcats should offer appreciable computation improvement’ on significantly larger datasets. To support this claim, I urge the authors to give some discussion of what their computational time complexity is for a user to get an idea of how the software will scale. (Also, Python could be less efficient in scaling up compared to Java due to its innate memory management.) How long did it take to run GOcats with the largest input they had so far? What if someone wants to run it on a larger sub-ontology such as ‘biological processes’?

Minor comments:

Again, this paper is well-presented. The following comments are mainly cosmetic changes that fix some small writing issues here and there:

1. In the Introduction, what do they mean by ‘(semi-) automated’? The term semi-automated is not well defined in this paper. I would rather be explicit about what manual procedures are needed, because GOcats could be interpreted as semi-automated, because it requires user-input (and perhaps user verification that the subgraph is indeed useful by visualization).

2. ‘Due to the nature of experimentally verified properties available, …’. sounds vague. What is the ‘nature’ of the ‘properties’?

3. ‘Due to the eventual application to the HPA datasets, …, were included to prevent categorization of terms that would complicate a eukaryotic interpretation…’. Were they ‘included’ or ‘excluded’? Why would including them prevent complications?

4. Typo ‘indeces’ occur in two places in the manuscript: ‘indeces’ -> ‘indices’

5. ‘Moreover, GOcats comparison with…’ sounds awkward grammatically. Consider rephrasing the whole sentence.

6. Punctuation ‘.’ Missing at the end of the legend of Figures 1C, 2, and 4.

Reviewer #2: The paper presents GOCats a novel tool that organizes the Gene Ontology (GO) into subgraphs representing user-defined concepts. This tool aims at mitigating the issues introduced by manual selection of higer-order GO terms to summarize results.

GOcats was evaluated using subcellular location categories to mine annotations from GO utilizing knowledgebases and evaluated their accuracy against immunohistochemistry datasets in the Human Protein Atlas (HPA) where it was shown to produce results comparable to mapping to GOslims and in some cases, potentially better results.

The tool addresses an important aspect. Many GO-based analyses suffer from manual selection of high-level GO terms to summarize results, which is time consuming and introduces potential bias.

However there are a number of issues that need to be clarified or improved upon.

1. The paper does not present results on how GOCats avoids the bias introduced by manual categorization. While it does eliminate the effort of manual work (just as M2S does), the authors themselves are aware of the potential for misuse.

2. Since GOCats are based on user input of keywords, the results for the analysis of the exact same data will very likely be different when done by two different researchers. This threatens reproducibility and comparison between studies. I would like to see the authors expand on this and on how their tool should be used to allow reproducibility. Please compare to GOSlims which are a shared model.

3. GOCats was developed for users not familiar with GO or bioinformatics. However, the correct usage of GOCats relies on user defining keywords at the right level of granularity (Figures 5 a and b illustrate this). If a user is not familiar with GO, how can they select an appropriate granularity level?

4. There are several references to the "traditionally problematic relation, has-part". Since GOCats has different usage modes "user has the option to define the scoping relation set." it would be really good to see the impact on results switching this on or off has.

5. There is a rather long review of semantic similarity and ontology mapping/evolution, themes that appear to be only marginally related to the topic. These portions of the text could be summarized.

6. A relevant application of GOCats is presented as "GOcats can facilitate the

integrity checking of annotations that are added to public repositories by streamlining the

process of extracting categories of annotations from knowledgebases and comparing

them to the original annotations in the raw data."

This needs to be explained in more detail. The workflow presented in the paper uses the raw data as the keyword input for GOCats, so how exactly would it result in an independent integrity checking is not clear to me.

Also this use case would be a lot stronger, if the reader was given an idea of how often cellular localization identification is not made to GO.

7. GOCats is potentially generalizable to the other GO types. Why was it only applied to cellular component? The other GO branches are much larger than CC, would this impact GOCats negatively?

8. In general, I find that the evaluation of the tool is lacking. I understand how it can potentially be useful, but the evaluation is based on a small controlled vocabulary in use by HPA. A better evaluation would be to run user studies, with users selecting keywords to categorize their data and reporting on their experience and usefulness of the results.

Minor:

9. This statement in Page 20 is unclear to me "Overall, the patterns of connectedness in this

network make more sense biologically, within the constraints of GO’s internal

organization." More sense compared to what?

10. In page 28 partial and no agreement definitions are not easy to understand in the text. Definition should not be in Figure 5 caption but in the main text.

11. In table 1 , how were the 25 examples selected?

6. PLOS authors have the option to publish the peer review history of their article (what does this mean?). If published, this will include your full peer review and any attached files.

Reviewer #1: No

Reviewer #2: No

---

## [Author Response · Author response to Decision Letter 0]

31 Mar 2020

Reviewer #1: 

The authors present GOcats, a tool that categorizes the Gene Ontology (GO) into subgraphs based on user inputs. Similar to GO-slim, it generates a sub-graph of the full GO using a list of user-provided keywords, while also handling the semantic scoping of the relationships within the GO. The paper itself only focuses on the sub-ontology under subcellular locations in order to be able report its accuracy against immunohistochemistry datasets in the Human Protein Atlas (HPA). Overall, it aims to reduce the manual effort required in hand-selecting a set of GO terms for categorization, which can improve the current complex workflows of GO analysis.

The paper is well-written, with an appreciable level of background and sufficient details for a scientist who is not completely familiar with the GO. The authors present how their work fits in among the numerous GO tools out there, and provide several examples of how the output of GOcats improve the correspondence between the annotations sources. The source code, scripts, and documentation are complete and appear up-to-date with the manuscript. The Methods section has included important details about their implementation, accompanied with reasonable explanation. For example, they have considered different user scenarios such as comprehensive and conservative subgraph extensions. As a statistician, I appreciate their discussion regarding how user-input keywords that prune the GO can increase the bias during enrichment analysis. A well-written software developed with the intent to maintain statistical reproducibility can also more generally benefit the field of data analysis. Overall, I would recommend this paper for publication, conditioned on the authors addressing some of my concerns below and some minor suggestions that could improve readability.

Response:

We thank the reviewer for their thorough review of our manuscript and recognizing the significance of our methods. We have addressed each of the reviewer’s comments below:

Issue 1:

Major concerns:

1. Distinction between this manuscript and their previous one [Hinderer EW, Flight RM, Dubey R, Macleod JN, Moseley HNB. Advances in gene ontology utilization improve statistical power of annotation enrichment. PLoS One. 2019;14(8):1–20.]

a. From what I understood, the previous manuscript also introduces GOcats as a tool that ‘organizes the Gene Ontology into subgraphs representing user-defined concepts, while ensuring that all appropriate relations are congruent with respect to scoping semantics.’ Is the underlying algorithm improved in any way? Was the previous software unable to perform the same analyses done in this paper?

Response:

In a perfect world, this manuscript would have been published first. However, the other paper was published first, because the significance of the application of GOcats was easier to perceive for other reviewers. This manuscript provides an in-depth description of GOcats’s methods and implementation and demonstrates different types of applications involving knowledgebase curation and deriving effective subcellular localization information from gene-product annotations. The other paper demonstrates the application of GOcats in annotation enrichment analysis, which is probably the most recognized use-case for GO at this time.

Issue 2:

b. At the end of the introduction, they mention ‘In a prior publication, we demonstrated GOcats’s ability to improve gene-annotation enrichment analyses, involving all GO sub-ontologies (23).’ I suggest that either (i) they should reference (23) earlier in the introduction to make it clear when it was initially developed, or (ii) they could discuss the differences between results from the previous paper and those in this paper. I skimmed the previous paper and indeed did not find any overlap, but I think it is the authors’ responsibly to clarify the differences to their previous work in this follow-up manuscript.

Response:

To better contrast this manuscript from the previous paper, we have added the following statements that clarify that this manuscript provides a thorough description of GOcats’s methods and implementation, along with their application in deriving gene-product-specific subcellular localization information and in knowledgebase curation:

“Due to the nature of the experimentally verified properties available from the HPA, our analysis in this paper focuses on cellular locations, especially subcellular locations. Also, this paper provides an in-depth description of GOcats’s methods and their implementation. In a prior publication, we demonstrated GOcats’s ability to improve gene-annotation enrichment analyses, involving all GO sub-ontologies (23).”

Issue 3:

2. Technical correctness of the algorithms. The pseudo-code in the Methods are interspersed and difficult to read without cross referencing. Here are some detailed pointers for the authors:

a. They mention ‘A single category representative root-node is selected by *recursively* counting the number of children each candidate term has and choosing the term with most children’. It is technically wrong to say the code block following the description recursive, because it iterative and does not consider the children of newly appended nodes.

Response:

The recursion is partly hidden in this example, because recursion is used to generate descendent sets. We stated this earlier in the manuscript as follows:

“At the first access of these sets through the ancestor or descendent property, the set is calculated with a recursive algorithm, stored for future use, and returned for immediate access.“

Part of the problem is deciding what level of detail to directly include in the pseudocode. To make the recursion clearer in this case, we have restated that the recursion takes place in the calculation of the descendents:

“A single category representative root-node is selected by recursively counting the number of children each candidate term has (i.e. creating the node.descendents) and choosing the term with the most children.”

Issue 4:

b. The object class definitions are inconsistent. I see both ‘FOR subnode in subgraph.nodes’ vs. ‘FOR subnode in subgraph’, and similar inconsistencies for the supergraph class in different code blocks. Although I couldn’t find their corresponding source code for these steps, I strongly suggest the authors to double-check their source and make the pseudo-code reflect their implementation more precisely.

Response:

We have made all of the pseudocode examples consistent.

Issue 5:

c. There are very long one-line codes such as ‘subnode.parent_node_set = …’ which are written in Python3 set-comprehension form. This is unnecessary for pseudo-code and makes it quite difficult parse the logic.

Response:

We reduced the font size and the amount of indentation to reduce the breakup of code statements across multiple lines. This improves the readability of these pseudocode blocks in the manuscript. We will work with the journal style editors to create an equivalent in the published form.

Issue 6:

3. Time complexity and scalability.

a.When comparing themselves with M2S, they say even though they were implemented in Python and M2S was implemented in Java, they were faster in performance due to ‘the utilization of stored ancestor and descendent node’. The statement is not convincing because M2S also stores the data but perhaps in a different format. They should be clear, for example, if it is because the node information is faster to access because they are in memory when the subgraph is built, or is it because they have a better way to parse the flat files that both M2S and they use. Right now, this discussion sounds too vague and mysterious to me. So instead of letting a reader guess what is happening, they should add just a couple more reasons accounting for any overhead that they and M2S do.

Response:

We have added the following statement to make this point clearer:

“However through the use of Python decorators, GOcats recursively creates and stores ancestor and descendent node sets in a manner analogous to lazy evaluation, allowing the implementation of efficient subgraph-centric algorithms that only precomputes the ancestor and descendent sets that are needed.”

Issue 7:

b.They say that ‘GOcats should offer appreciable computation improvement’ on significantly larger datasets. To support this claim, I urge the authors to give some discussion of what their computational time complexity is for a user to get an idea of how the software will scale. (Also, Python could be less efficient in scaling up compared to Java due to its innate memory management.) How long did it take to run GOcats with the largest input they had so far? What if someone wants to run it on a larger sub-ontology such as ‘biological processes’?

Response:

In the other GOcats paper, we demonstrate the use of GOcats on all three GO sub-ontologies, including biological process. GOcats generates these results for all three GO sub-ontologies in a few seconds. We believe that the precomputation and storage of the ancestor and descendent sets has complexity O(n log n); however, since GOcats runs on all of GO in just seconds, we have not felt the need to rigorously test the computational complexity of GOcats’s algorithms. We have added the following statement to support our point that GOcats performs very efficiently on all of GO:

“Based on these results, GOcats should offer appreciable computational improvement on significantly larger datasets. This is demonstrated in GOcats’s application in annotation enrichment analysis involving all three GO sub-ontologies, which executes in just a few seconds (23).”

Issue 8:

Minor comments:

Again, this paper is well-presented. The following comments are mainly cosmetic changes that fix some small writing issues here and there:

1. In the Introduction, what do they mean by ‘(semi-) automated’? The term semi-automated is not well defined in this paper. I would rather be explicit about what manual procedures are needed, because GOcats could be interpreted as semi-automated, because it requires user-input (and perhaps user verification that the subgraph is indeed useful by visualization).

Response:

We mean at least partially automated by the term (semi-)automated. And we do consider several use-cases of GOcats to be semi-automated due to the requirement for user input. We have added a clarifying phrase to the Introduction:

“Therefore, (semi-)automated (i.e. at least partially automated) and unbiased methods for categorizing semantically-similar and biologically-related annotations are needed for integrating information from heterogeneous sources—even if the annotation terms themselves are standardized—to facilitate effective downstream systems-level analyses and integrated network-based modeling.”

Issue 9:

2. ‘Due to the nature of experimentally verified properties available, …’. sounds vague. What is the ‘nature’ of the ‘properties’?

Response:

We have added the clarifying phrase “from the HPA” into this sentence:

“Due to the nature of the experimentally verified properties available from the HPA, our analysis in this paper focuses on cellular locations, especially subcellular locations.”

Issue 10:

3. ‘Due to the eventual application to the HPA datasets, …, were included to prevent categorization of terms that would complicate a eukaryotic interpretation…’. Were they ‘included’ or ‘excluded’? Why would including them prevent complications?

Response:

Thank you again! We sometimes forget to explicitly spell out all of the logical steps in our arguments. In this instance, we are considering the implications of a greedy subgraph extension algorithm. We have tried to make this easier to follow with the following revision:

“Due to the eventual application to the HPA datasets, three unusual categories, “bacterial”, “viral”, and “other organism”, were included to prevent categorization of terms that would complicate a eukaryotic interpretation of the other 22 subcellular locations, within the context of a greedy subgraph extension algorithm. “ 

Issue 11:

4. Typo ‘indeces’ occur in two places in the manuscript: ‘indeces’ -> ‘indices’

Response:

Fixed.

Issue 12:

5. ‘Moreover, GOcats comparison with…’ sounds awkward grammatically. Consider rephrasing the whole sentence.

Response:

We rephrased it as follows:

“Moreover, the comparison of GOcats to M2S demonstrates similar mapping performance between the two methods, but with GOcats providing important improvements in mapping, computational speed, ease of use, and flexibility of use.”

Issue 13:

6. Punctuation ‘.’ Missing at the end of the legend of Figures 1C, 2, and 4.

Response:

Fixed.

Reviewer #2: 

The paper presents GOCats a novel tool that organizes the Gene Ontology (GO) into subgraphs representing user-defined concepts. This tool aims at mitigating the issues introduced by manual selection of higer-order GO terms to summarize results.

GOcats was evaluated using subcellular location categories to mine annotations from GO utilizing knowledgebases and evaluated their accuracy against immunohistochemistry datasets in the Human Protein Atlas (HPA) where it was shown to produce results comparable to mapping to GOslims and in some cases, potentially better results.

The tool addresses an important aspect. Many GO-based analyses suffer from manual selection of high-level GO terms to summarize results, which is time consuming and introduces potential bias.

However there are a number of issues that need to be clarified or improved upon.

Response:

We thank the reviewer for their review of our manuscript. We have addressed each of the reviewer’s comments below:

Issue 1:

1. The paper does not present results on how GOCats avoids the bias introduced by manual categorization. While it does eliminate the effort of manual work (just as M2S does), the authors themselves are aware of the potential for misuse.

Response:

There are two different sources of bias that are mentioned in this manuscript. GOcats provides an automated way to build subgraph categories. This eliminates potential bias that can come from the manual building of these subgraph categories. We have the following statement in the manuscript, highlighting this point:

“But because GOcats will always produce the same subgraph categorizations for the same set of keywords used with the same version of GO, we argue that our categorization is more reproducible and less prone to bias than manually grouping GO terms into categories or otherwise manually identifying major concepts represented from omics-level analyses.”

However, GOcats is still prone to bias that comes from any user input, which in this case, is the keywords and terms provided by the user. We have tried to be careful and clearly indicate what biases GOcats can and cannot avoid.

Issue 2:

2. Since GOCats are based on user input of keywords, the results for the analysis of the exact same data will very likely be different when done by two different researchers. This threatens reproducibility and comparison between studies. I would like to see the authors expand on this and on how their tool should be used to allow reproducibility. Please compare to GOSlims which are a shared model.

Response:

This is an issue of reproducibility. Using the same keywords with the same versions of GO and datasets will produce the same results. We have a public FigShare repository that includes all of the manuscript’s results and the programs and scripts used to produce these results. One researcher simply needs to provide the keywords they used along with the versions of GOcats, GO, and their datasets for another researcher to reanalyze. This is no different than a shared GO slim. Also, GO slims do change over time. We have added the following statement to highlight the point of enhanced reproducibility:

“Furthermore, the set of keywords can be provided along with the version of GOcats, GO, and the dataset to enable reproducibility of analyses by others.”

Issue 3:

3. GOCats was developed for users not familiar with GO or bioinformatics. However, the correct usage of GOCats relies on user defining keywords at the right level of granularity (Figures 5 a and b illustrate this). If a user is not familiar with GO, how can they select an appropriate granularity level?

Response:

GOcats has multiple use-cases. We describe how it can be used to generate GO slim like categories, deriving subcellular location from a large knowledgebase, and knowledgebase curation. In another publication, we demonstrates GOcats’s use in annotation enrichment analysis. These use-cases require different levels of GO and bioinformatics expertise. Dataset harmonization and knowledgebase curation where granularity adjustment would be useful would require quite a bit of expertise. We have tried to illustrate this with the following added statement:

“Furthermore, GOcats was designed for scientists who are less familiar with GO; however, the package has advanced features for users with more bioinformatics expertise.“ 

Issue 4:

4. There are several references to the "traditionally problematic relation, has-part". Since GOCats has different usage modes "user has the option to define the scoping relation set." it would be really good to see the impact on results switching this on or off has.

Response:

Our other publication on GOcats provides results illustrating the effect of turning on and off the has_part relationship in categorization and annotation enrichment analysis.

Issue 5:

5. There is a rather long review of semantic similarity and ontology mapping/evolution, themes that appear to be only marginally related to the topic. These portions of the text could be summarized.

Response:

We have found it necessary to provide an introduction to these topics so that those unfamiliar with them can understand the significance of our work. A lot of people use GO for a wide range of purposes and have quite a bit of expertise in specific applications of GO. Therefore, a lot of people view themselves as “experts” on GO and on ontologies as a whole; however, they have little formal training in ontologies as an area of research. Therefore, we find ourselves needing to provide the necessary background in order for others to understand exactly what problems we are solving and why the solutions are significant. 

Issue 6:

6. A relevant application of GOCats is presented as "GOcats can facilitate the integrity checking of annotations that are added to public repositories by streamlining the process of extracting categories of annotations from knowledgebases and comparing them to the original annotations in the raw data."

This needs to be explained in more detail. The workflow presented in the paper uses the raw data as the keyword input for GOCats, so how exactly would it result in an independent integrity checking is not clear to me.

Also this use case would be a lot stronger, if the reader was given an idea of how often cellular localization identification is not made to GO.

Response:

The word “raw” is a misnomer. We meant to contrast the raw HPA datasets with respect to derived ontology-normalized information stored in a knowledgebase. The raw HPA datasets are actually highly processed results that used a controlled vocabulary to describe specific subcellular locations. To make this point clearer, we have added the following statement:

‘In this context, the term “raw data” refers to processed, curated experimental data that is annotated as a contrast to the GO annotations derived from a knowledgebase.’

In response to the reviewer’s comment about an independent integrity check, we directly compared the subcellular localization indicated by the “raw” HPA datasets to the HPA-deposited annotations in the knowledgebase. We hope this is clearer by the statement we add above.

With respect to reviewer’s request the we provide an idea of how often cellular localization identification is not made with GO, we cannot feasibly review all potential uses of GO that have been published to give an idea of how often this occurs. However, we have pointed out a major instance when cellular compartments were manually organized into a hierarchical localization tree:

‘For example, a recent effort to create a protein-protein interaction network analysis database resorted to manually building a hierarchical localization tree from GO cellular compartment terms due to the “incongruity in the resolution of localization data” in various source databases and the fact that no published method existed at that time for the automated organization of such terms (6).’ 

Issue 7:

7. GOCats is potentially generalizable to the other GO types. Why was it only applied to cellular component? The other GO branches are much larger than CC, would this impact GOCats negatively?

Response:

Our other GOcats publication illustrates GOcats applied to all three GO sub-ontologies.

Issue 8:

8. In general, I find that the evaluation of the tool is lacking. I understand how it can potentially be useful, but the evaluation is based on a small controlled vocabulary in use by HPA. A better evaluation would be to run user studies, with users selecting keywords to categorize their data and reporting on their experience and usefulness of the results.

Response:

GOcats is a versatile tool. Our other publication illustrates GOcats use in annotation enrichment analyses of RNAseq datasets. This manuscript is meant to provide a thorough description of the methods and implementation with interesting applications in categorization and curation. We have provided a thorough evaluation of these use-cases. Evaluation of the usability of GOcats by end-users is not the point of this manuscript. Demonstration of new capabilities in automated categorization methods is the point of this manuscript. We have done this. 

Issue 9:

Minor:

9. This statement in Page 20 is unclear to me "Overall, the patterns of connectedness in this

network make more sense biologically, within the constraints of GO’s internal

organization." More sense compared to what?

Response:

We have added the following clarifying statement:

“In other words, it is easier to see the expected biological relationships between cellular locations in Figure 1B versus Figure 1A.”

Issue 10:

10. In page 28 partial and no agreement definitions are not easy to understand in the text. Definition should not be in Figure 5 caption but in the main text.

Response:

Respectfully, we disagree. It is painful for the reader to have to flip back and forth from Figure and main text to understand what each column means. However, we have added these definitions to the main text as well:

‘In this context, “partial agreement” refers to genes with at least one matching subcellular location, “partial agreement is superset” refers to genes where knowledgebase subcellular locations are a superset of the HPA dataset (these are mutually exclusive to the “partial agreement” category), "no agreement" refers to genes with no subcellular locations in common, and “no annotations” refers to genes in the experimental dataset that were not found in the knowledgebase.’

Issue 11:

11. In table 1 , how were the 25 examples selected?

Response:

As we stated in the text:

‘Starting with common biological subcellular concepts like “nucleus”, “cytoplasm”, and “mitochondrion”, we recursively used terms not being categorized to identify additional subcellular concepts and associated keywords represented within the GO Cellular Component sub-ontology.’

---

## [Decision Letter · Decision Letter 1]

4 May 2020

GOcats: A tool for categorizing Gene Ontology into subgraphs of user-defined concepts

PONE-D-20-00699R1

Dear Dr. Moseley,

We are pleased to inform you that your manuscript has been judged scientifically suitable for publication and will be formally accepted for publication once it complies with all outstanding technical requirements.

With kind regards,

Marc Robinson-Rechavi

Academic Editor

PLOS ONE

Additional Editor Comments (optional):

Reviewers' comments:

Reviewer's Responses to Questions

**Comments to the Author**

1. If the authors have adequately addressed your comments raised in a previous round of review and you feel that this manuscript is now acceptable for publication, you may indicate that here to bypass the “Comments to the Author” section, enter your conflict of interest statement in the “Confidential to Editor” section, and submit your "Accept" recommendation.

Reviewer #1: All comments have been addressed

Reviewer #3: All comments have been addressed

2. Is the manuscript technically sound, and do the data support the conclusions?

Reviewer #1: Yes

Reviewer #3: Yes

3. Has the statistical analysis been performed appropriately and rigorously? 

Reviewer #1: N/A

Reviewer #3: N/A

4. Have the authors made all data underlying the findings in their manuscript fully available?

Reviewer #1: Yes

Reviewer #3: Yes

5. Is the manuscript presented in an intelligible fashion and written in standard English?

Reviewer #1: Yes

Reviewer #3: Yes

6. Review Comments to the Author

Reviewer #1: The authors have addressed my concerns and fixed the places that I was confused about. Please make sure that the figures are readable in the final print.

Reviewer #3: ... Please use the space provided to explain your answers to the questions above.

Thank you for addressing my comments.

7. PLOS authors have the option to publish the peer review history of their article (what does this mean?). If published, this will include your full peer review and any attached files.

Reviewer #1: No

Reviewer #3: Yes: Pascale Gaudet

---

## [Editor Report · Acceptance letter]

28 May 2020

PONE-D-20-00699R1 

GOcats: A tool for categorizing Gene Ontology into subgraphs of user-defined concepts 

Dear Dr. Moseley:

I am pleased to inform you that your manuscript has been deemed suitable for publication in PLOS ONE. Congratulations! Your manuscript is now with our production department. 

With kind regards,

on behalf of

Prof. Marc Robinson-Rechavi 

Academic Editor

PLOS ONE